# Enhancing Parallelism in Decentralized Stochastic Convex Optimization

**Ofri Eisen** [* 1]  **Ron Dorfman** [* 1]  **Kfir Y. Levy** [1]

## Abstract

Decentralized learning has emerged as a powerful approach for handling large datasets across multiple machines in a communication-efficient manner. However, such methods often face scalability limitations, as increasing the number of machines beyond a certain point negatively impacts convergence rates. In this work, we propose *Decentralized Anytime SGD*, a novel decentralized learning algorithm that significantly extends the critical parallelism threshold, enabling the effective use of more machines without compromising performance. Within the stochastic convex optimization (SCO) framework, we establish a theoretical upper bound on parallelism that surpasses the current state-of-the-art, allowing larger networks to achieve favorable statistical guarantees and closing the gap with centralized learning in highly connected topologies.

## 1. Introduction

Distributed learning has become a key paradigm for large-scale machine learning (ML), where multiple machines train an ML model by utilizing their collective data and computational resources (Verbraeken et al., 2020). This approach enables efficient ML scaling by accelerating training through parallel computation. Distributed ML systems are especially useful when data is inherently distributed across multiple users, as in federated learning (McMahan et al., 2017; Kairouz et al., 2021), where preserving local data privacy is a priority.

Distributed learning systems are broadly categorized into two primary architectures: *centralized* and *decentralized*. Centralized systems rely on a central parameter server (PS) to coordinate training by aggregating local computations and distributing global model updates (Li et al., 2014). While this design simplifies training management, it faces scalability challenges due to communication bottlenecks and is inherently vulnerable to a single point of failure since the server is essential to system operation.

In contrast, decentralized systems rely on direct interactions between neighboring machines, eliminating the need for a central PS (Kempe et al., 2003; Boyd et al., 2006; Nedic & Ozdaglar, 2009; Lian et al., 2017). While this architecture introduces challenges in coordination and maintaining global model consistency, it offers significant advantages. By reducing the risk of a single point of failure and enhancing communication efficiency, decentralized systems provide greater flexibility and resilience (Assran et al., 2019; Kong et al., 2021; Yuan et al., 2021; Li et al., 2022).

A central trade-off in distributed learning, both centralized and decentralized, lies in balancing parallelism with statistical efficiency. While increasing parallelism—by incorporating more machines—can accelerate training, it may degrade learning efficiency beyond a certain point. This issue is particularly acute in decentralized systems, where sparse communication topologies amplify performance degradation, limiting the effective utilization of additional machines.

Existing decentralized methods reveal a persistent gap between centralized and decentralized learning in their ability to scale with the number of machines (Koloskova et al., 2020; 2021; He et al., 2022). Surprisingly, this gap is observed even in highly connected network topologies, where information between nodes propagates faster, effectively mimicking the behavior of centralized systems. This raises the question of whether decentralized methods are fundamentally constrained in fully exploiting increased parallelism, even under ideal communication conditions.

In this work, we propose *Decentralized Anytime SGD (DAT-SGD)*, a novel and simple algorithm designed to enhance parallelism in decentralized systems. It relies on the Anytime SGD framework (Cutkosky, 2019), where stochastic gradient descent (SGD) updates are performed using gradients evaluated at *averaged* iterates. Each machine performs these updates locally and exchanges information with its neighbors. By relying on averaged query points that evolve more slowly than the iterates themselves, our approach effectively mitigates the bias caused by local model inconsistencies, also known as consensus distance.

---

[*]Equal contribution [1]Department of Electrical and Computer Engineering, Technion, Haifa, Israel. Correspondence to: Ofri Eisen <ofri.eisen@campus.technion.ac.il>.

*Proceedings of the 42nd International Conference on Machine Learning*, Vancouver, Canada. PMLR 267, 2025. Copyright 2025 by the author(s).

*Table 1.* **Parallelism Bounds.** Comparison of parallelism bounds for decentralized SCO across different network topologies for Decentralized SGD (D-SGD, Koloskova et al., 2020) and our method, expressed in terms of the total number of samples $N = M \cdot T$. For reference, the centralized case achieves a bound of $\mathcal{O}(\sqrt{N})$. NC stands for near-complete graph. Higher parallelism bounds are better, allowing us to accelerate the learning process without degrading performance.

| TOPOLOGY | $1/\rho$ | D-SGD | DAT-SGD |
|---|---|---|---|
| RING | $\mathcal{O}(M^2)$ | $\mathcal{O}(N^{1/8})$ | $\mathcal{O}(N^{1/6})$ |
| TORUS | $\mathcal{O}(M)$ | $\mathcal{O}(N^{1/6})$ | $\mathcal{O}(N^{1/4})$ |
| NC | $\approx 1$ | $\mathcal{O}(\rho^{1/2}N^{1/4})$ | $\mathcal{O}(\rho\sqrt{N})$ |

We perform our analysis within the stochastic convex optimization (SCO) framework—a powerful framework which captures several classical problems like SVMs and linear regression, as well as serving as a testbed towards designing and analyzing ML algorithms.

For convex and smooth functions, our method achieves an error of $\epsilon$ over a decentralized network of $M$ machines in $T = \mathcal{O}(1/M\epsilon^2 + 1/\rho\epsilon)$ iterations, where $\rho \in (0,1]$ is the *spectral gap* of the network, a parameter that captures the connectivity of the network topology (see Definition 2.5 for a formal definition). This result implies that **(1)** for complete (or near-complete) topologies, where $\rho = \Omega(1)$, our algorithm recovers the convergence rate of centralized methods (Dekel et al., 2012), thus filling the gap implied by prior methods; and **(2)** for general topologies, it improves upon the previously best-known rate of $\mathcal{O}(1/M\epsilon^2 + 1/\rho^{1/2}\epsilon^{3/2})$ (Koloskova et al., 2020), thereby enabling larger networks while maintaining linear speed-up; see Table 1 for a comparison of parallelism bounds for common network topologies. In Section 2.1, we elaborate on the computation of these parallelism bounds.

## 1.1. Related Work

Centralized learning systems are widely adopted in both industry and academia, thanks to their simplicity and the effectiveness of Minibatch SGD (Dekel et al., 2012), which is the most widespread approach to training in such systems. While alternative centralized methods like local-SGD have been explored in recent years (McMahan et al., 2017; Kairouz et al., 2021), Minibatch SGD still serves as a cornerstone in large-scale training due to its ease of implementation and strong empirical performance.

Despite the enduring popularity of centralized training, decentralized systems present an appealing alternative by eliminating the need for a central parameter server and enabling direct communication among neighboring machines (Tsitsiklis, 1984; Kempe et al., 2003; Boyd et al., 2006; Nedic

& Ozdaglar, 2009; Lian et al., 2017). This design not only mitigates the risk of a single point of failure but also offers greater scalability and flexibility (Assran et al., 2019; Kong et al., 2021; Yuan et al., 2021). A common mechanism underpinning these benefits is the *gossip averaging* protocol (Xiao & Boyd, 2004; Boyd et al., 2006), a low-overhead communication method that efficiently disseminates model updates across the network.

In the context of stochastic optimization, decentralized training approaches have garnered considerable interest in recent years. The stochastic convex case was explored by Lian et al. (2017), who analyzed the Decentralized-SGD (D-SGD) method. Later, Koloskova et al. (2020) extended this approach within a unified framework that accounts for changing topologies and randomized gossip communication. While these aforementioned methods degrade in the face of data heterogeneity, Koloskova et al. (2021) completely eliminated this issue by employing an approach called gradient-tracking (Di Lorenzo & Scutari, 2016; Nedic et al., 2017; Pu & Nedić, 2021). See Table 2 for a comparison of convergence rates.

Curiously, all existing methods present convergence bounds that imply a clear gap between the centralized and decentralized cases, even for highly connected topologies. Concretely, in centralized systems one may use up to $\mathcal{O}(\sqrt{N})$ machines to accelerate the learning process without degrading generalization (Dekel et al., 2012), where $N$ is the total number of samples used in the process. Conversely, in decentralized systems the best known parallelization limit is $M \leq \mathcal{O}((\rho\sqrt{N})^{1/2})$, which is substantially smaller. See Table 1 for bounds regarding different topologies.

Decentralized training has also been extensively studied in the context of stochastic non-convex optimization. This was done in (Lian et al., 2017; Koloskova et al., 2020; 2021), which have derived analogous bounds to the ones they achieved in the convex setting. Additionally, Kong et al. (2021) empirically studied the interplay between the local model parameters and gossip communication. More recently, He et al. (2022) improved convergence rates for non-convex problems by replacing local SGD updates with local momentum updates, establishing a parallelization limit of $M \leq \mathcal{O}((\rho\sqrt{N})^{2/3})$. However, this remains below the centralized bound of $M \leq \mathcal{O}(\sqrt{N})$, which applies in the non-convex setting.

To further improve the parallelism bound, we build on a recent technique that involves gradually shifting query points in SCO (Cutkosky, 2019). This approach has been leveraged in recent works to improve asynchronous (Aviv et al., 2021) and local (Dahan & Levy, 2024) training methods, as well as to design universal accelerated algorithms (Kavis et al., 2019; Ene et al., 2021; Antonakopoulos et al., 2022).

*Table 2.* **Convergence to $\epsilon$-accuracy.** Comparison of convergence rates with prior work in convex decentralized learning. Here, $\sigma$ denotes noise variance, $\zeta$ represents data heterogeneity, and $\rho$ is the spectral gap. We note that the leading term $\sigma^2/M\epsilon^2$ is statistically optimal (Nemirovski & Yudin, 1983).

| REFERENCE | CONVERGENCE RATE |
|---|---|
| KOLOSKOVA ET AL. (2020) | $\mathcal{O}\left( \frac{\sigma^2}{M\epsilon^2} + \frac{\sigma\sqrt{\rho}+\zeta}{\rho\epsilon^{3/2}} + \frac{1}{\rho\epsilon} \right)$ |
| KOLOSKOVA ET AL. (2021) | $\mathcal{O}\left( \frac{\sigma^2}{M\epsilon^2} + \frac{\sigma}{\sqrt{\rho}\epsilon^{3/2}} + \frac{1}{\rho\epsilon} \right)$ |
| THIS WORK | $\mathcal{O}\left( \frac{\sigma^2}{M\epsilon^2} + \frac{\sqrt{\sigma}+\sqrt{\zeta}}{\rho\epsilon} + \frac{1}{\epsilon} \right)$ |

## 2. Problem Setup and Background

In this section, we formally define our problem setup and provide an overview of relevant background.

We study decentralized SCO problems, where the objective is to minimize a convex loss function $f : \mathbb{R}^d \to \mathbb{R}$ defined as follows:

$$f(x) := \frac{1}{M} \sum_{i=1}^{M} \{f_i(x) := \mathbb{E}_{z\sim\mathcal{D}_i}[f_i(x, z)]\} \,, \quad (1)$$

where $f_i : \mathbb{R}^d \to \mathbb{R}$ represents the loss function on machine $i \in \{1, \ldots, M\}$, $\mathcal{D}_i$ is the local data distribution, and $f_i(\cdot, z)$ is the instance-dependent loss for a given sample $z$ on machine $i$.

We focus on first-order optimization methods, where in round $t$, each machine $i$ samples $z_t^i \sim \mathcal{D}_i$, computes a stochastic gradient estimate $\nabla f_i(x_t^i, z_t^i)$, and uses it to update its local iterate $x_t^i$. The method ultimately produces an output $x_{\text{output}}$, and its performance is measured by the expected *excess loss*, defined as:

$$\mathbb{E}[\text{Excess-Loss}] := \mathbb{E}[f(x_{\text{output}})] - f(x^*) \,,$$

where $x^* \in \arg\min_{x\in\mathbb{R}^d} f(x)$ is a minimizer of $f$, and the expectation is taken over the randomness of the samples.

Throughout, we make the following standard assumptions.

**Assumption 2.1** (Smoothness). Each function $f_i$ is $L$-smooth, i.e., for any $x, y \in \mathbb{R}^d$ we have:

$$f_i(y) \le f_i(x) + \nabla f_i(x)^T(y - x) + \frac{L}{2}\|y - x\|^2 \,.$$

**Assumption 2.2** (Bounded noise variance). There exists a constant $\sigma^2$ such that for all $x \in \mathbb{R}^d$ and $i \in [M]$:

$$\mathbb{E}_{z\sim\mathcal{D}_i}\|\nabla f_i(x, z) - \nabla f_i(x)\|^2 \le \sigma^2 \,.$$

**Assumption 2.3** (Bounded heterogeneity). There exists a constant $\zeta^2$ such that for all $x \in \mathbb{R}^d$:

$$\frac{1}{M} \sum_{i=1}^{M} \|\nabla f_i(x) - \nabla f(x)\|^2 \le \zeta^2 \,.$$

We note that our bounded heterogeneity assumption has been widely adopted in prior works (Lian et al., 2017; Tang et al., 2018a; Li et al., 2019; He et al., 2022). Notably, some studies have explored milder assumptions, requiring bounded heterogeneity only at the optimum (Koloskova et al., 2020) or at the origin (Lu & De Sa, 2021), or even removing this assumption entirely by leveraging techniques such as gradient tracking (Nedic et al., 2017; Koloskova et al., 2021). In this work, we adopt the standard bounded heterogeneity assumption for simplicity, leaving potential relaxations to future work.

**Decentralized Training.** We consider systems (networks) where nodes are connected through a communication graph, and each node can efficiently communicate with its neighbors. In such decentralized systems, nodes represent machines, and edges correspond to communication links.

*Gossip averaging* is a robust communication protocol that efficiently and reliably propagates information across the decentralized network without relying on a central coordinator (Xiao & Boyd, 2004; Boyd et al., 2006; Nedic & Ozdaglar, 2009; Shi et al., 2014; Koloskova et al., 2020).

Concretely, suppose each machine $i$ in the network holds a vector $x_0^i \in \mathbb{R}^d$, and our goal is to efficiently and robustly communicate the *consensus* $\bar{x} := \frac{1}{M} \sum_i x_0^i$ to all of the machines in the network. Gossip averaging is a sequential process towards doing so, where each machine computes a sequence $\{x_t^i\}_t$ such that $x_t^i$ eventually converges to $\bar{x}$ for all $i$. At every gossip step, each node $i$ updates its vector as a weighted average of its neighbors:

$$x_{t+1}^i = \sum_{j=1}^{M} P_{ij} x_t^j \,,$$

where $P_{ij} > 0$ if nodes $i$ and $j$ are connected, and $P \in [0, 1]^{M\times M}$ is a given gossip matrix which satisfies the following properties.

**Definition 2.4** (Gossip matrix). A *gossip matrix $P \in [0, 1]^{M\times M}$* is a symmetric, doubly stochastic matrix. That is, $P$ satisfies $P = P^T$ as well as $P\mathbf{1} = \mathbf{1}$ and $\mathbf{1}^T P = \mathbf{1}^T$.

These properties imply that **(i)** for any node $i$ then $\{P_{ij}\}_{j\in[M]}$ is a weight vector (i.e., has positive entries which sum to 1), and that **(ii)** the consensus is preserved i.e. that $\forall t, \bar{x} = \frac{1}{M} \sum_{i=1}^{M} x_0^i = \frac{1}{M} \sum_{i=1}^{M} x_t^i$.

As is standard in the literature we shall assume that the matrix $P$ is given and satisfies the above properties.

Note that gossip averaging can be written in matrix form as:

$$X_{t+1} = X_t P \,,$$

where $X_t := \begin{pmatrix} x_t^1 & x_t^2 & \cdots & x_t^M \end{pmatrix} \in \mathbb{R}^{d\times M}$.

A key characteristic of the gossip matrix is the *spectral gap*, reflecting the connectivity degree of the network topology.

**Definition 2.5** (Spectral gap). For a gossip matrix $P$ with eigenvalues $1 = \lambda_1 > |\lambda_2| \geq \cdots \geq |\lambda_M|$, the *spectral gap* is defined as:

$$\rho := 1 - |\lambda_2| \in (0, 1].$$

An important property of gossip averaging is its contractive effect on the distance from the average of local vectors.

**Property 2.6.** Let $X \in \mathbb{R}^{d \times M}$ and define $\bar{X} := X \frac{1}{M} \mathbf{1} \mathbf{1}^\top$. Let $P \in \mathbb{R}^{M \times M}$ be a gossip matrix with spectral gap $\rho$. Then,

$$\|XP - \bar{X}\|_F^2 \leq (1 - \rho)\|X - \bar{X}\|_F^2 . \tag{2}$$

The above, along with the consensus-preserving property, ensures that iterative gossip averaging converges to the consensus (i.e., the average of the local vectors) exponentially fast. Notably, in complete graph topologies where $\rho = 1$, consensus is achieved in a single gossip step, effectively mimicking centralized behavior.

### 2.1. Parallelism Bounds

It has been well established in the literature that the convergence rate of SGD for convex functions is given by $\mathcal{O}(\sigma/\sqrt{T} + 1/T)$, where $T$ is the number of iterations (Nemirovski et al., 2009; Agarwal et al., 2012). Assuming each iteration processes a single sample, we have $N = T$ total samples.

In distributed systems, computation is distributed across $M$ machines, therefore accelerating the training process. With $M$ machines, we process $N$ samples in just $T = N/M$ iterations. However, as we mentioned, this parallelization introduces a trade-off, as increasing $M$ can eventually reduce efficiency, as we show next.

For centralized systems, Dekel et al. (2012) derived the following convergence rate for Mini-batch (Parallel) SGD:

$$\mathbb{E}[\text{Excess-Loss}] = \mathcal{O}\left(\frac{\sigma}{\sqrt{MT}} + \frac{1}{T}\right) = \mathcal{O}\left(\frac{\sigma}{\sqrt{N}} + \frac{M}{N}\right).$$

Thus, as long as $M \leq \mathcal{O}(\sqrt{N})$ (ignoring $\sigma$), we can increase $M$ without degrading performance.

For decentralized systems, Koloskova et al. (2020) analyzed the Decentralized SGD (D-SGD) algorithm, obtaining the error rate:

$$\mathcal{O}\left(\frac{\sigma}{\sqrt{MT}} + \frac{\sigma^{2/3}\rho^{1/3} + \zeta^{2/3}}{\rho^{2/3}T^{2/3}} + \frac{1}{\rho T}\right) =$$
$$\mathcal{O}\left(\frac{\sigma}{\sqrt{N}} + \frac{\left(\sigma^{2/3}\rho^{1/3} + \zeta^{2/3}\right)M^{2/3}}{\rho^{2/3}N^{2/3}} + \frac{M}{\rho N}\right) . \tag{3}$$

Interestingly, the parallelism bound differs based on data heterogeneity. In homogeneous setups ($\zeta = 0$), we can scale up to $M \leq \mathcal{O}(\rho^{1/2}N^{1/4})$, whereas in heterogeneous settings ($\zeta > 0$), the limit tightens to $M \leq \mathcal{O}(\rho N^{1/4})$, which can be significantly lower for sparse topologies. It is worth mentioning that Koloskova et al. (2021) enhanced this bound, by incorporating a gradient tracking mechanism, eliminating the dependence on $\zeta$ altogether.

Note that even for dense graph topologies, where $\rho \approx 1$, the rate in (3) does not match the centralized case. As we establish in Section 4, our approach bridges this gap, achieving an error rate of $\mathcal{O}(\sigma/\sqrt{MT} + (\sqrt{\sigma} + \sqrt{\zeta})/\rho T + 1/T)$. This improvement increases the parallelism bound to $M \leq \mathcal{O}(\rho\sqrt{N})$. Table 2 provides a comparison of convergence rates with prior work.

## 3. The Pitfall of Decentralized SGD

Next, we discuss the D-SGD algorithm and outline its key limitation. Given the structure of decentralized topologies, the D-SGD algorithm naturally extends SGD. As described in Algorithm 1, each machine alternates between updating its local model weights using the standard SGD rule and exchanging information with its neighbors via gossip averaging. However, unlike in centralized Minibatch SGD, where all machines compute gradients at the same query points, decentralized training lacks immediate synchronization. Since gossip-based communication does not instantly enforce consensus (unless $\rho = 1$), local models diverge, and each machine evolves independently.

This lack of synchronization introduces a fundamental challenge: **gradient estimates at each node become biased with respect to the global consensus**, affecting convergence analysis. As shown in (Koloskova et al., 2019), this bias is closely tied to the consensus distance $\Xi_t := \frac{1}{M}\sum_{i=1}^{M}\|w_t^i - \bar{w}_t\|^2$, where $\bar{w}_t = \frac{1}{M}\sum_{i=1}^{M}w_t^i$ represents the global average of local models. The consensus distance quantifies the discrepancy between local models, and limited communication makes it challenging to minimize. This issue can be framed as a competition between the rate of consensus achievement and the rate at which local parameters evolve. If all machines synchronized instantly, D-SGD would closely resemble centralized Minibatch SGD. However, for any $\rho < 1$, perfect synchronization remains unattainable, necessitating to control and bound this bias.

To gain intuition, consider the D-SGD update rule under the simplifying assumption that the gradient variance across nodes is bounded: $\Psi_t := \frac{1}{M}\sum_{i=1}^{M}\|g_t^i - \bar{g}_t\|^2 \leq G^2$ for all $t \geq 1$, where $\bar{g}_t = \frac{1}{M}\sum_{i=1}^{M}g_t^i$ is the global average of local stochastic gradients.

Using standard gossip averaging analysis and the SGD update rule, we can obtain the following recursion for the

**Algorithm 1** Decentralized SGD (D-SGD)

**Input:** Initial point $w_1 \in \mathbb{R}^d$, learning rate $\eta$, gossip matrix $P$, number of rounds $T$.
Initialize $w_1^i = w_1$ for all $i \in [M]$
**for** $t = 1, \ldots, T$ **do**
  **for** $i \in [M]$ in parallel **do**
    Sample $z_t^i \sim \mathcal{D}$ and set $g_t^i = \nabla f_i(w_t^i, z_t^i)$
    $w_{t+\frac{1}{2}}^i = w_t^i - \eta g_t^i$
    $w_{t+1}^i = \sum_{j=1}^M w_{t+\frac{1}{2}}^j P_{ij}$
  **end for**
**end for**

consensus distance:

$$\Xi_{t+1} = \frac{1}{M} \sum_{i=1}^M \|w_{t+1}^i - \bar{w}_{t+1}\|^2$$

$$\leq \frac{1-\rho}{M} \sum_{i=1}^M \|w_{t+\frac{1}{2}}^i - \bar{w}_{t+\frac{1}{2}}\|^2$$

$$= \frac{1-\rho}{M} \sum_{i=1}^M \|(w_t - \eta g_t^i) - (\bar{w}_t - \eta \bar{g}_t)\|^2$$

$$\leq \left(1 - \frac{\rho}{2}\right) \Xi_t + C \cdot \frac{G^2 \eta^2}{\rho}, \qquad (4)$$

for some constant $C > 0$. Solving this recursion yields the bound $\Xi_t \leq \mathcal{O}(\eta^2/\rho^2)$ for all $t \geq 1$. In practice, Koloskova et al. (2020) conducted a more refined analysis, obtaining an improved bound of order $\mathcal{O}(\eta^2/\rho)$. This leads to an overall error rate of $\mathcal{O}(1/\eta T + \eta + \eta^2/\rho)$. Optimizing the learning rate by balancing these terms suggests choosing $\eta \lesssim (\rho/T)^{1/3}$, which in turn results in the $\mathcal{O}(1/\rho^{1/3}T^{2/3})$ term in (3), ultimately limiting the parallelism bound. Ideally, if we could tighten the bound on the consensus distance, it would lead to a direct improvement in this term. In the next section, we achieve this by introducing gradually shifting anchor points.

## 4. DAT-SGD: Decentralized Anytime SGD

In this section, we introduce DAT-SGD. We begin in Section 4.1 by discussing the Anytime SGD method, which serves as the foundation of our approach. Then, in Section 4.2, we extend this framework to the decentralized setting. Finally, in Section 4.3, we provide an intuitive analysis and a proof sketch of the convergence statement, highlighting how Anytime SGD enables to mitigate the consensus distance—an idea we further elaborate on in Section 4.4.

### 4.1. Anytime SGD

For this part, consider a single-machine setup. The widely studied SGD method generates a sequence of iterates $\{w_t\}_t$

**Algorithm 2** Decentralized Anytime SGD (DAT-SGD)

1: **Input:** Initial iterate $w_1$, learning rate $\eta$, gossip matrix $P$, number of rounds $T$, non-negative weights $\{\alpha_t\}_{t \geq 1}$.
2: Initialize $w_1^i = x_1^i = w_1$ for all $i \in [M]$
3: **for** $t = 1, \ldots, T$ **do**
4:   **for** $i \in [M]$ in parallel **do**
5:     Sample $z_t^i \sim \mathcal{D}_i$ and set $g_t^i = \nabla f_i(x_t^i, z_t^i)$
6:     Local updates
7:     $w_{t+\frac{1}{2}}^i = w_t^i - \eta \alpha_t g_t^i$
8:     $x_{t+\frac{1}{2}}^i = \frac{\alpha_{1:t-1}}{\alpha_{1:t}} x_t^i + \frac{\alpha_t}{\alpha_{1:t}} w_{t+\frac{1}{2}}^i$
9:     Gossip communication
10:     $w_{t+1}^i = \sum_{j=1}^M w_{t+\frac{1}{2}}^j P_{ij}$
11:     $x_{t+1}^i = \sum_{j=1}^M x_{t+\frac{1}{2}}^j P_{ij}$
12:   **end for**
13: **end for**

using the update rule $w_{t+1} = w_t - \eta g_t$, where $g_t$ is a stochastic gradient estimate computed at the current iterate $w_t$. In contrast, the Anytime SGD framework (Cutkosky, 2019) computes stochastic gradients at different query points—specifically, the weighted average of past iterates.

Formally, given a sequence of non-negative weights $\{\alpha_t\}_t$, Anytime SGD generates two sequences, $\{w_t\}_t$ and $\{x_t\}_t$, according to the update rules:

$$w_{t+1} = w_t - \eta \alpha_t g_t, \qquad (5)$$

$$x_{t+1} = \frac{\alpha_{1:t-1}}{\alpha_{1:t}} x_t + \frac{\alpha_t}{\alpha_{1:t}} w_{t+1}, \qquad (6)$$

where $x_1 = w_1$, $\alpha_{1:t} \coloneqq \sum_{\tau=1}^t \alpha_\tau$ for all $t > 0$, and $\alpha_{1:0} \coloneqq 0$. Here, $g_t$ is an estimate of the gradient at the *weighted average* $x_t$.

Cutkosky (2019) introduced the Anytime framework to ensure that the query points converge to the optimal solution—unlike standard SGD, where iterates may not. It was shown that Anytime SGD achieves the same convergence rates as SGD for convex functions. However, the averaged query points $\{x_t\}_t$ exhibit greater stability, changing slower than the iterates themselves. As we show next, this approach also enables establishing a last-iterate convergence guarantee.

### 4.2. Extension to the Decentralized Setup

Our approach extends Anytime SGD to the decentralized setting, as outlined in Algorithm 2. In round $t$, each machine performs the updates given in Equations (5) and (6) locally and, similar to D-SGD, shares both its model weights, $w_t^i$, and query points, $x_t^i$, through gossip averaging.

The convergence of Algorithm 2 is established in Theorem 4.1, with the full proof deferred to Appendix B.

**Theorem 4.1.** *Under Assumptions 2.1-2.3, consider Algorithm 2 with linear weights $\alpha_t = t$ and a learning rate given by*

$$\eta = \min \left\{ \frac{1}{24LT}, \frac{\rho^2}{K}, \frac{D_1\sqrt{M}}{\sqrt{3}\sigma T^{3/2}}, \sqrt{\frac{D_1}{2K\tilde{\sigma}}} \frac{\rho}{T} \right\} ,$$

*where $D_1^2 := \|w_1 - x^*\|^2$, $K^2 := 5120L^2$, and $\tilde{\sigma}^2 := 2\sigma^2 + \zeta^2$. Then, for all $T \geq 1$, the following bound holds:*

$$\mathbb{E}[f(\bar{x}_T) - f^*] \leq \mathcal{O}\left( \frac{\sigma D_1}{\sqrt{MT}} + \frac{D_1^{3/2}\sqrt{L\tilde{\sigma}}}{\rho T} + \frac{LD_1^2}{T} \right) ,$$

*where $\bar{x}_T := \frac{1}{M} \sum_{i=1}^{M} x_T^i$.*

Let $N = MT$ be the total number of samples after $T$ iterations. From Theorem 4.1, ignoring dependencies on $L$, $D_1$, $\sigma$ and $\zeta$, we obtain the convergence rate $\mathcal{O}(1/\sqrt{N} + M/\rho N)$. This implies that the asymptotic upper bound on parallelism is $\mathcal{O}(\rho\sqrt{N})$. Beyond this, the second term of $M/\rho N$ dominates, leading to a decline in learning efficiency when increasing the number of machines. This bound represents a significant improvement over the prior $\mathcal{O}(\rho^{1/2}N^{1/4})$ parallelism limit (Koloskova et al., 2020; 2021).

In addition, we can establish the convergence of the local iterates, as follows. The proof is provided in Appendix B.1.

**Corollary 4.2.** *Under the same assumptions and parameter selections as in Theorem 4.1, the local iterate at each machine $i \in \{1, \ldots, M\}$ satisfies:*

$$\mathbb{E}[f(x_T^i) - f^*] \leq$$
$$\mathcal{O}\left( \frac{\sigma D_1}{\sqrt{MT}} + \frac{D_1^{3/2}\sqrt{L\tilde{\sigma}}}{\rho T} + \frac{LD_1^2}{T} + \frac{M\tilde{\sigma}D_1}{\rho^2 T^2} \right) .$$

*The additional last term does not affect the established upper bound on parallelism, which remains $M \leq \mathcal{O}(\rho\sqrt{N})$.*

**Transient Iteration Complexity.** A complementary metric to the parallelism bound is the transient iteration complexity, which quantifies how many iterations are needed before the convergence rate matches that of centralized SGD, i.e., when the $\sigma/\sqrt{MT}$ term dominates (Pu et al., 2021). From Theorem 4.1, it follows that the transient iteration complexity of our method is $\mathcal{O}(M/\rho^2)$, representing an improvement over D-SGD by a factor of $M^2$. For instance, this complexity corresponds to $\mathcal{O}(M^5)$ for the ring topology and $\tilde{\mathcal{O}}(M)$ for the exponential graph (Ying et al., 2021).

### 4.3. Proof Sketch

To complement the analysis in Section 3, we first provide an intuitive analysis of the consensus distance. As formally shown in Appendices A and B, the bias in Anytime SGD is

related to the average consensus distance between the query points, defined as

$$\Gamma_t := \frac{1}{M} \sum_{i=1}^{M} \|x_t^i - \bar{x}_t\|^2, \quad \text{where} \quad \bar{x}_t = \frac{1}{M} \sum_{i=1}^{M} x_t^i .$$

Since the query points evolve more gradually than the iterates, we can derive a tighter bound on $\Gamma_t$. To illustrate this, we consider a simplified setting with uniform weights $\alpha_t = 1$ for all $t$ and assume $\Psi_t \leq G^2$ for all $t \geq 1$, as in Section 3. Using gossip averaging for the query points and the Anytime averaging, we can show that:

$$\Gamma_{t+1} = \frac{1}{M} \sum_{i=1}^{M} \|x_{t+1}^i - \bar{x}_{t+1}\|^2$$

$$\leq \frac{1-\rho}{M} \sum_{i=1}^{M} \|x_{t+\frac{1}{2}}^i - \bar{x}_{t+\frac{1}{2}}\|^2$$

$$= \frac{1-\rho}{M} \sum_{i=1}^{M} \left\| \left(1 - \frac{1}{t}\right)(x_t^i - \bar{x}_t) + \frac{1}{t}(w_{t+\frac{1}{2}}^i - \bar{w}_{t+\frac{1}{2}}) \right\|^2$$

$$\leq \left(1 - \frac{\rho}{2}\right)\Gamma_t + \frac{C}{\rho t^2}\left(\Xi_t + G^2\eta^2\right) . \tag{7}$$

for some constant $C > 0$. Recall that $\Xi_t$ satisfies its own recursion, as derived in Equation (4). Using the refined bound $\Xi_t \leq \mathcal{O}(G^2\eta^2/\rho)$, we obtain:

$$\Gamma_{t+1} \lesssim \left(1 - \frac{\rho}{2}\right)\Gamma_t + \frac{C}{\rho t^2}\left(\frac{G^2\eta^2}{\rho} + G^2\eta^2\right)$$

$$\leq \left(1 - \frac{\rho}{2}\right)\Gamma_t + 2C \cdot \frac{G^2\eta^2}{\rho^2 t^2} . \tag{8}$$

Summing over $t \in [T]$ and rearranging terms, we can get the bound $\sum_{t=1}^{T} \Gamma_t \leq \mathcal{O}(\eta^2/\rho^3)$. Since the final error rate is proportional to the average consensus distance over time, $\frac{1}{T}\sum_{t=1}^{T}\Gamma_t$, the resulting error is of order $\mathcal{O}(1/\eta T + \eta + \eta^2/\rho^3 T)$. Tuning the learning rate by setting $\eta \lesssim \rho$ yields the $\mathcal{O}(1/\rho T)$ term, which enables enhanced parallelism.

This simplified analysis provides intuition on how slowly changing query points improve the rate. Next, we present a more formal proof sketch capturing finer transition details.

*Proof Sketch of Theorem 4.1.* As we show in Lemma B.1, the consensus query point sequence $\{\bar{x}_t\}_{t \geq 1}$ is an $\{\alpha_t\}_{t \geq 1}$-weighted average of the consensus iterates sequence $\{\bar{w}_t\}_{t \geq 1}$. Moreover, the average iterates sequence evolves similarly to Equation (5), following the update rule $\bar{w}_{t+1} = \bar{w}_t - \eta\alpha_t\bar{g}_t$. Thus, the consensus sequences $\{\bar{x}_t\}_{t \geq 1}$ and $\{\bar{w}_t\}_{t \geq 1}$ align with the structure of Anytime SGD, allowing us to leverage standard results applicable to Anytime SGD. Specifically, defining $\Delta_t := \mathbb{E}[f(\bar{x}_t) - f^*]$, it follows for all $t \geq 1$ that (cf. Lemma A.3):

$$\alpha_{1:t}\Delta_t \leq \frac{D_1^2}{\eta} + \eta\sum_{\tau=1}^{T}\alpha_\tau^2\mathbb{E}\|\bar{g}_\tau\|^2 + 4\eta T \cdot B_T , \tag{9}$$

where $B_T := \mathbb{E}\left[\sum_{\tau=1}^{T} \alpha_\tau^2 \|\mathbb{E}\bar{g}_\tau - \nabla f(\bar{x}_\tau)\|^2\right]$. Focusing on the middle term, we show using standard arguments that:

$$\mathbb{E}\|\bar{g}_\tau\|^2 \leq \frac{3\sigma^2}{M} + 3\mathbb{E}\|\mathbb{E}\bar{g}_\tau - \nabla f(\bar{x}_\tau)\|^2 + 3\mathbb{E}\|\nabla f(\bar{x}_\tau)\|^2 .$$

Plugging this into Equation (9) allows us to derive the following bound:

$$\alpha_{1:t}\Delta_t \leq \frac{D_1^2}{\eta} + \frac{3\sigma^2\eta}{M}\sum_{\tau=1}^{T}\alpha_\tau^2 + 8\eta T B_T$$

$$+ 3\eta\sum_{\tau=1}^{T}\alpha_\tau^2\mathbb{E}\|\nabla f(\bar{x}_\tau)\|^2 . \quad (10)$$

A key component in our proof is bounding $B_T$, which is essentially the sum of weighted squared biases. As we show in Section 4.4, this term is bounded as follows:

$$B_T \leq \frac{K^2\tilde{\sigma}^2\eta^2}{2\rho^4}\sum_{\tau=1}^{T}\alpha_\tau^2 .$$

Therefore, from Equation (10), we get:

$$\alpha_{1:t}\Delta_t \leq \frac{D_1^2}{\eta} + \frac{3\sigma^2\eta}{M}\sum_{\tau=1}^{T}\alpha_\tau^2 + \frac{4K^2\tilde{\sigma}^2\eta^3 T}{\rho^4}\sum_{\tau=1}^{T}\alpha_\tau^2$$

$$+ 3\eta\sum_{\tau=1}^{T}\alpha_\tau^2\mathbb{E}\|\nabla f(\bar{x}_\tau)\|^2 . \quad (11)$$

For the last sum, the smoothness of $f$ implies that $\|\nabla f(x)\|^2 \leq 2L(f(x) - f^*)$ for all $x \in \mathbb{R}^d$, yielding:

$$\sum_{\tau=1}^{T}\alpha_\tau^2\mathbb{E}\|\nabla f(\bar{x}_\tau)\|^2 \leq 2L\sum_{\tau=1}^{T}\alpha_\tau^2\Delta_\tau \leq 4L\sum_{\tau=1}^{T}\alpha_{1:\tau}\Delta_\tau ,$$

where the last inequality follows from $\alpha_\tau^2 = \tau^2 \leq 2\alpha_{1:\tau}$. Substituting this bound into (11) and recalling that $\eta \leq \frac{1}{24LT}$, we obtain:

$$\alpha_{1:t}\Delta_t \leq \frac{D_1^2}{\eta} + \frac{3\sigma^2\eta}{M}\sum_{\tau=1}^{T}\alpha_\tau^2 + \frac{4K^2\tilde{\sigma}^2\eta^3 T}{\rho^4}\sum_{\tau=1}^{T}\alpha_\tau^2$$

$$+ \frac{1}{2T}\sum_{\tau=1}^{T}\alpha_{1:\tau}\Delta_\tau . \quad (12)$$

Observe that $a_t := \alpha_{1:t}\Delta_t$ follows the structure $a_t \leq b + \frac{1}{2T}\sum_{\tau=1}^{T}a_\tau, \ \forall t \in [T]$, which implies that $a_t \leq 2b$ for all $t \in [T]$ (see Lemma D.3). In particular, for $t = T$, we get:

$$\alpha_{1:T}\Delta_T \leq \frac{2D_1^2}{\eta} + \frac{6\sigma^2\eta}{M}\sum_{\tau=1}^{T}\alpha_\tau^2 + \frac{8K^2\tilde{\sigma}^2\eta^3 T}{\rho^4}\sum_{\tau=1}^{T}\alpha_\tau^2 .$$

Dividing by $\alpha_{1:T}$, noting that $\frac{\sum_{\tau=1}^{T}\alpha_\tau^2}{\alpha_{1:T}} \leq T$ for linear weights, and appropriately tuning the learning rate concludes the proof. $\square$

## 4.4. Bounding the Bias

A central part of our analysis is bounding the bias introduced by local model differences. The bias bound is formally stated in Lemma 4.3, and the complete proof can be found in Appendix B.2. A brief proof sketch is provided below.

**Lemma 4.3.** *Consider the setting in Theorem 4.1 and let $\bar{g}_t := \frac{1}{M}\sum_{i=1}^{M}g_t^i$ and $\bar{x}_t := \frac{1}{M}\sum_{i=1}^{M}x_t^i$ denote the average gradient and query point across machines at round $t$, respectively. Then, for linear weights $\alpha_t = t$, and a learning rate bounded as $\eta \leq \frac{\rho^2}{K}$, the following bound holds:*

$$B_T := \mathbb{E}\left[\sum_{\tau=1}^{T}\alpha_\tau^2\|\mathbb{E}\bar{g}_\tau - \nabla f(\bar{x}_\tau)\|^2\right] \leq \frac{K^2\tilde{\sigma}^2\eta^2}{2\rho^4}\sum_{\tau=1}^{T}\alpha_\tau^2.$$

*where we recall that $K^2 := 5120L^2$ and $\tilde{\sigma}^2 := 2\sigma^2 + \zeta^2$.*

*Proof Sketch.* By Jensen's inequality and the smoothness of each $f_i$, we can bound each term in the sum as follows:

$$\mathbb{E}\|\mathbb{E}\bar{g}_\tau - \nabla f(\bar{x}_\tau)\|^2 \leq \frac{1}{M}\sum_{i=1}^{M}\mathbb{E}\|\nabla f_i(x_\tau^i) - \nabla f_i(\bar{x}_\tau)\|^2$$

$$\leq \frac{L^2}{M}\sum_{i=1}^{M}\mathbb{E}\|x_\tau^i - \bar{x}_\tau\|^2 .$$

As previously defined, let $\Gamma_t := \frac{1}{M}\sum_{i=1}^{M}\mathbb{E}\|x_t^i - \bar{x}_t\|^2$ denote the average consensus distance of the query points at round $t$. Thus,

$$B_T \leq L^2\sum_{\tau=1}^{T}\alpha_\tau^2\Gamma_\tau . \quad (13)$$

Our objective is therefore to bound $\Gamma_\tau$ for every $\tau \in [T]$. Using standard gossip averaging analysis and the query points averaging, we derive the following recursion for $\Gamma_t$:

$$\Gamma_{t+1} \leq \left(1 - \frac{\rho}{2}\right)\Gamma_t + \frac{4\delta_t^2}{\rho}\left(\Xi_t + \eta^2\alpha_t^2\Psi_t\right) , \quad (14)$$

recalling that $\Xi_t := \frac{1}{M}\sum_{i=1}^{M}\mathbb{E}\|w_t^i - \bar{w}_t\|^2$ and $\Psi_t := \frac{1}{M}\sum_{i=1}^{M}\mathbb{E}\|g_t^i - \bar{g}_t\|^2$ represent the average consensus distances of the iterates and gradients at round $t$, respectively, and using $\delta_t := \frac{\alpha_t}{\alpha_{1:t}}$. Notably, $\Xi_t$ satisfies its own recursion (see Lemma C.4):

$$\Xi_{t+1} \leq \left(1 - \frac{\rho}{2}\right)\Xi_t + \frac{2\eta^2\alpha_t^2}{\rho}\Psi_t .$$

Solving this recursion yields the following for all $t \in [T]$:

$$\Xi_t \leq \frac{2\eta^2\alpha_t^2}{\rho}\sum_{\tau=1}^{t-1}\left(1 - \frac{\rho}{2}\right)^{t-1-\tau}\Psi_\tau .$$

Substituting this back into (14) gives:

$$\Gamma_{t+1} \leq \left(1 - \frac{\rho}{2}\right)\Gamma_t$$
$$+ \frac{4\eta^2\alpha_t^2\delta_t^2}{\rho}\left(\Psi_t + \frac{2}{\rho}\sum_{\tau=1}^{t-1}\left(1 - \frac{\rho}{2}\right)^{t-1-\tau}\Psi_\tau\right).$$

In Lemma C.2, we show that $\Psi_t \leq 5\tilde{\sigma}^2 + 10L^2\Gamma_t$ for all $t \in [T]$. Plugging this bound, noting that for linear weights $\alpha_t^2\delta_t^2 = \mathcal{O}(1)$, and using some algebra, we obtain:

$$\Gamma_{t+1} \leq \left(1 - \kappa + \frac{c_1^2\eta^2}{\kappa}\right)\Gamma_t$$
$$+ \frac{c_1^2\eta^2}{\kappa^2}\sum_{\tau=1}^{t-1}(1-\kappa)^{t-1-\tau}\Gamma_\tau + \frac{c_2^2\eta^2}{\kappa^3},$$

where $\kappa = \frac{\rho}{2}$, $c_1^2 = \Theta(K^2)$, and $c_2^2 = \Theta(\tilde{\sigma}^2)$. We have now derived a recursion for $\Gamma_t$ that no longer depends on $\Xi_t$ or $\Psi_t$. For a sufficiently small learning rate, specifically $\eta \leq \frac{\rho^2}{K}$, this recursion can be explicitly solved, yielding:

$$\Gamma_t \leq \frac{2c_2^2\eta^2}{\kappa^4} = \Theta\left(\frac{\tilde{\sigma}^2\eta^2}{\rho^4}\right).$$

Injecting this bound into (13) yields the result:

$$B_T \leq L^2\sum_{\tau=1}^{T}\alpha_\tau^2\Gamma_\tau = \Theta\left(\frac{K^2\tilde{\sigma}^2\eta^2}{\rho^4}\sum_{\tau=1}^{T}\alpha_\tau^2\right).$$

$\square$

# 5. Experiments

In this section, we empirically evaluate our method on both a synthetic least squares problem and an image classification task. All experiments are conducted using three random seeds, and we report the averaged results.

## 5.1. Least Squares on Synthetic Data

We begin with a synthetic least squares problem to illustrate key theoretical properties of our algorithm. For each machine, the local objective function is defined as $f_i(x) = \frac{1}{2}\|A_i x - b_i\|^2$, where $A_i \in \mathbb{R}^{d \times d}$ is drawn from a standard multivariate normal distribution. The targets vector is set as $b_i = A_i(x^\sharp - \delta_i)$, where $x^\sharp \sim \mathcal{N}(0, \frac{1}{d}I_d)$ is sampled once per configuration, and $\delta_i \sim \mathcal{N}(0, \frac{\zeta^2}{d}I_d)$ introduces heterogeneity across machines.[1] To incorporate stochasticity, we add Gaussian noise $\xi \sim \mathcal{N}(0, \frac{\sigma^2}{d}I_d)$ when querying local gradients, resulting in the noisy gradient estimate $\nabla f_i(x) + \xi$. In our experiments, we set $d = 50$.

We compare our method with D-SGD, evaluating performance across three network topologies: ring, torus,

---

[1] Here, $\zeta^2$ quantifies heterogeneity **at the optimum**.

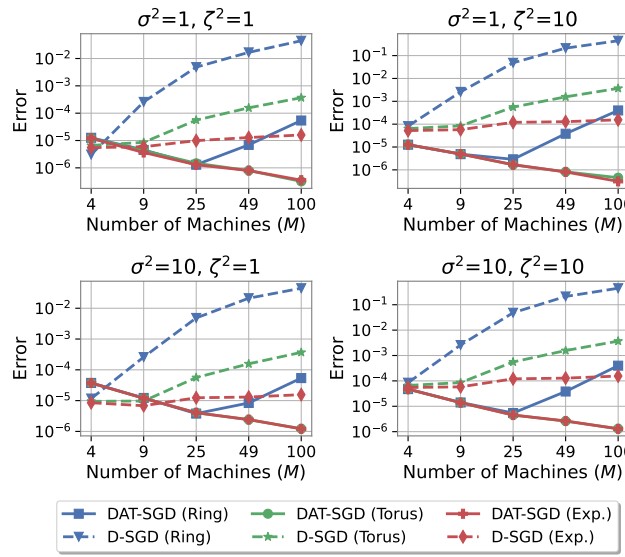

*Figure 1.* Final error on synthetic least squares problem for different numbers of machines and various gradient noise variance ($\sigma^2$) and heterogeneity ($\zeta^2$) levels over ring, torus, and exponential graph topologies. We plot $\frac{1}{M}\sum_{i=1}^{M}\|x_T^i - x^*\|^2$ and $\frac{1}{M}\sum_{i=1}^{M}\|w_T^i - x^*\|^2$ for DAT-SGD and D-SGD, respectively.

and 1-peer exponential graph (Ying et al., 2021). The exponential graph is a fast-mixing topology for which $1/\rho = \mathcal{O}(\log M)$. For each method and topology, we perform a grid search over the learning rate $\eta \in \{0.0001, 0.0005, 0.001, 0.005, 0.01, 0.05, 0.1\}$ and select the value that yields the lowest error after 100K iterations. For DAT-SGD, we use constant weights $\alpha_t = 1$ for all $t$.

In Figure 1, we plot the final error as a function of the number of machines, across four configurations defined by $\sigma, \zeta \in \{1, 10\}$, with different colors indicating the underlying topology. For D-SGD, we observe that performance degrades on the ring and torus topologies starting from $M = 4$ and $M = 9$, respectively, while it remains relatively stable on the 1-peer exponential graph. In contrast, our method improves with larger $M$: performance steadily improves on the torus and exponential graphs, and performs better on the ring topology up to $M = 25$ before deteriorating. This suggests that, beyond a certain threshold (between $M = 25$ and 49), the network-dependent error term (which scales as $\mathcal{O}(1/\rho T) = \mathcal{O}(M^2/T)$ for the ring) becomes dominant. Overall, the results align with our theoretical findings, as DAT-SGD enables performance improvement for larger $M$. We provide complete convergence curves in Appendix E.

## 5.2. Image Classification with a Neural Network

Next, we evaluate our method on the Fashion MNIST (Xiao et al., 2017) image classification task using the LeNet (LeCun et al., 1998) architecture. The data is partitioned among

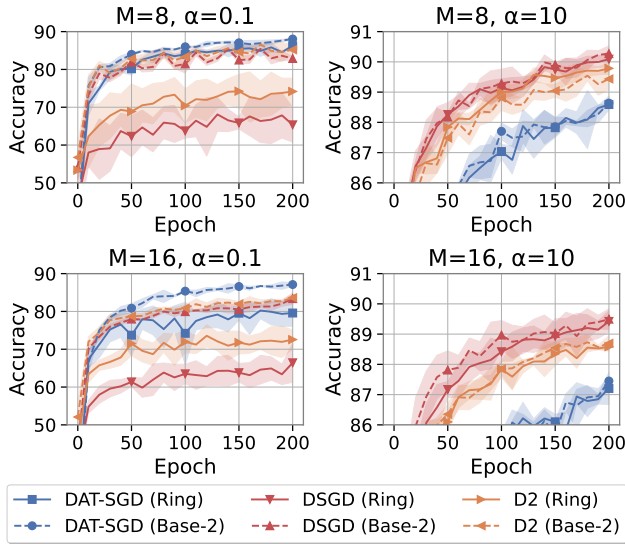

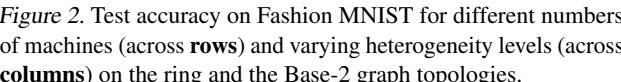

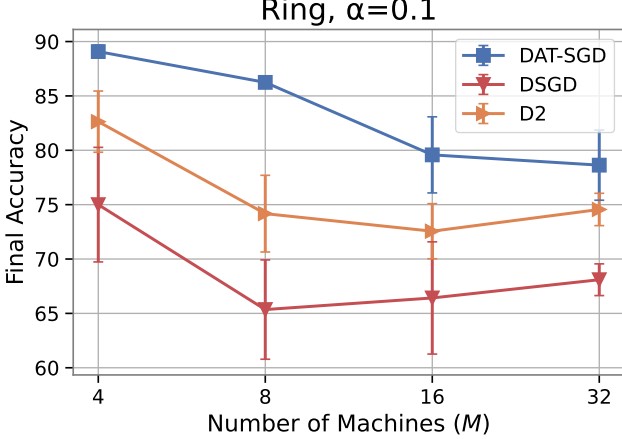

Figure 3. Final test accuracy on Fashion MNIST for varying number of machines on the ring topology with heterogeneous data.

Figure 2. Test accuracy on Fashion MNIST for different numbers of machines (across **rows**) and varying heterogeneity levels (across **columns**) on the ring and the Base-2 graph topologies.

workers following a Dirichlet distribution with parameter $\alpha$, which controls the heterogeneity level (Hsu et al., 2019).

We compare our method against D-SGD and $D^2$ (Tang et al., 2018b), a decentralized optimization method specifically designed to improve robustness to data heterogeneity. Experiments are performed on the ring topology and the Base-2 Graph (Takezawa et al., 2023)—a time-varying, state-of-the-art topology for decentralized learning. For our method and D-SGD, we use momentum with parameter $\beta = 0.9$. For each method and topology, the learning rate was selected via grid search over $\eta \in \{0.001, 0.01, 0.1\}$.

Unlike the synthetic least squares problem, this task is non-convex. Following the heuristic proposed by Dahan & Levy (2025), we adopt a momentum-like update for Anytime averaging of the form $x_{t+1} = \gamma_t x_t + (1 - \gamma_t) w_t$, using a fixed value of $\gamma_t = 0.9$. This choice was shown to enhance training stability and adaptability in non-convex landscapes.

Figure 2 depicts the test accuracy for $M = 8$ and 16 machines under heterogeneous ($\alpha = 0.1$) and nearly homogeneous ($\alpha = 10$) data, with different colors indicating the compared methods. In the heterogeneous case, our method consistently outperforms the baselines across both topologies, with results on the ring matching (for $M = 8$) or nearly matching (for $M = 16$) those of the baselines on the well-connected Base-2 Graph. Conversely, and perhaps unexpectedly, under homogeneous data, D-SGD and $D^2$ achieve better performance, motivating further investigation into this gap across data regimes in non-convex settings. In Figure 3, similar to Figure 1, we show the final accuracy for different numbers of machines on the ring topology with

heterogeneous data ($\alpha = 0.1$). Notably, the largest accuracy drop for DAT-SGD occurs between $M = 8$ and 16, whereas D-SGD and $D^2$ degrade most between $M = 4$ and 8, demonstrating our claim of improved parallelism.

## 6. Conclusion and Future Work

In this work, we presented DAT-SGD, a simple yet powerful approach to decentralized SCO that raises the parallelism threshold, enabling to increase the number of machines in the network while maintaining linear speed-up. This is achieved by effectively mitigating the consensus distance using the Anytime SGD mechanism, which computes stochastic gradients at gradually changing query points, thereby limiting local model divergence.

Several promising directions emerge from our work. One is integrating our method with gradient tracking, which has been shown to remove dependence on data heterogeneity (Koloskova et al., 2021). Moreover, we assume the gossip matrix is symmetric and doubly stochastic, allowing us to use Property 2.6 for a clear and simple analysis. Extending our results to asymmetric or row/column stochastic matrices, as in, e.g., (Assran et al., 2019), remains an open problem. Finally, establishing convergence bounds in the non-convex setting is a compelling challenge for future research.

## Acknowledgments

We thank the reviewers for their valuable comments. This research was partially supported by Israel PBC-VATAT, by the Technion Artificial Intelligence Hub (Tech.AI), and by the Israel Science Foundation (grant No. 3109/24).

## Impact Statement

This paper presents work whose goal is to advance the field of Machine Learning. There are many potential societal consequences of our work, none of which we feel must be specifically highlighted here.

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

# A. Anytime SGD Analysis

In this section, we discuss key properties of Anytime SGD with biased gradients. The main result is Lemma A.3, which will later be adapted to the decentralized, gossip-based communication setup in Appendix B

For an initial iterate $w_1 \in \mathbb{R}^d$, learning rate $\eta > 0$, and non-negative weights $\{\alpha_t\}_{t\geq 1}$, we consider the sequences $\{w_t\}_{t\geq 1}$ and $\{x_t\}_{t\geq 1}$, defined by the update rules:

$$w_{t+1} = w_t - \eta\alpha_t g_t, \tag{15}$$

$$x_{t+1} = \frac{\alpha_{1:t-1}}{\alpha_{1:t}}x_t + \frac{\alpha_t}{\alpha_{1:t}}w_{t+1}, \quad \text{with} \quad x_1 = w_1 . \tag{16}$$

Here, $g_t$ is a biased estimate of the gradient of some function $f$ at $x_t$.

The following result mimics the gradient inequality for the $\{\alpha_t\}_{t\geq 1}$-weighted averages.

**Lemma A.1** (Cutkosky, 2019; Theorem 1, Dahan & Levy, 2024). *Let $f : \mathbb{R}^d \to \mathbb{R}$ be a convex function with global minimum $f^* := f(x^*)$, and let the sequence $\{x_t\}_{t\geq 1}$ be defined as in Equation (16). Then, for any $t \geq 1$, the following holds:*

$$0 \leq \alpha_{1:t}\left(f(x_t) - f^*\right) \leq \sum_{\tau=1}^{t} \alpha_\tau \nabla f(x_\tau)^\top (w_\tau - x^*) .$$

Next, we present a regret bound for online gradient descent with weighted gradients.

**Lemma A.2** (Cutkosky, 2019; Lemma 2, Dahan & Levy, 2024). *Consider the update rule in Equation (15). Then, for all $t \geq 1$, it holds that:*

$$\sum_{\tau=1}^{t} \alpha_\tau g_\tau^\top (w_\tau - x^*) \leq \frac{\|w_1 - x^*\|^2}{2\eta} + \frac{\eta}{2}\sum_{\tau=1}^{t} \alpha_\tau^2 \|g_\tau\|^2 .$$

The following result is fundamental to our analysis and plays a key role in establishing our main proof. The analysis closely follows that of Dahan & Levy (2024) and is included here for completeness; see Appendix F therein for further details.

**Lemma A.3.** *For the sequences $\{w_t\}_{t\geq 1}$ and $\{x\}_{t\geq 1}$ defined in Equations (15) and (16), the following holds for any $t \geq 1$:*

$$\alpha_{1:t}\mathbb{E}\left[f(x_t) - f^*\right] \leq \frac{\|w_1 - x^*\|^2}{\eta} + \eta\sum_{\tau=1}^{T} \alpha_\tau^2 \mathbb{E}\|g_\tau\|^2 + 4\eta T\sum_{\tau=1}^{T} \alpha_\tau^2 \mathbb{E}\|\mathbb{E}g_\tau - \nabla f(x_\tau)\|^2 .$$

*Proof.* Lemma A.1 implies the following:

$$\alpha_{1:t}\mathbb{E}\left[f(x_t) - f^*\right] \leq \mathbb{E}\left[\sum_{\tau=1}^{t} \alpha_\tau \nabla f(x_\tau)^\top (w_t - x^*)\right]$$

$$= \mathbb{E}\left[\sum_{\tau=1}^{t} \alpha_\tau g_\tau^\top (w_\tau - x^*)\right] + \mathbb{E}\left[\sum_{\tau=1}^{t} \alpha_\tau \left(\nabla f(x_\tau) - g_\tau\right)^\top (w_\tau - x^*)\right]$$

$$\leq \frac{\|w_1 - x^*\|^2}{2\eta} + \frac{\eta}{2}\sum_{\tau=1}^{t} \alpha_\tau^2 \mathbb{E}\|g_\tau\|^2 + \sum_{\tau=1}^{t} \mathbb{E}\left[\alpha_\tau \left(\nabla f(x_\tau) - g_\tau\right)^\top (w_\tau - x^*)\right], \tag{17}$$

where the last inequality follows from Lemma A.2. Every element in the rightmost sum can be bounded using Cauchy-Schwarz inequality and Young's inequality, $a \cdot b \leq \frac{1}{2\theta}\|a\|^2 + \frac{\theta}{2}\|b\|^2$, which holds for any $\theta > 0$, yielding:

$$\mathbb{E}\left[\alpha_\tau \left(\nabla f(x_\tau) - g_\tau\right)^\top (w_\tau - x^*)\right] = \mathbb{E}\left[\alpha_\tau \left(\nabla f(x_\tau) - \mathbb{E}g_\tau\right)^\top (w_\tau - x^*)\right]$$

$$\leq \mathbb{E}\left[\alpha_\tau \|\mathbb{E}g_\tau - \nabla f(x_\tau)\| \cdot \|w_\tau - x^*\|\right]$$

$$\leq \frac{\alpha_\tau^2}{2\theta}\mathbb{E}\|\mathbb{E}g_\tau - \nabla f(x_\tau)\|^2 + \frac{\theta}{2}\mathbb{E}\|w_\tau - x^*\|^2 , \tag{18}$$

where the first equality follows from the law of total expectation.

Substituting (18) into (17) gives:

$$\alpha_{1:t}\left(f(x_t) - f^*\right) \leq \frac{\|w_1 - x^*\|^2}{2\eta} + \frac{\eta}{2}\sum_{\tau=1}^{t}\alpha_\tau^2\mathbb{E}\|g_\tau\|^2 + \frac{1}{2\theta}\sum_{\tau=1}^{t}\alpha_\tau^2\mathbb{E}\|\mathbb{E}g_\tau - \nabla f(x_\tau)\|^2 + \frac{\theta}{2}\sum_{\tau=1}^{t}\mathbb{E}\|w_\tau - x^*\|^2$$

$$\leq \frac{\|w_1 - x^*\|^2}{2\eta} + \frac{\eta}{2}\sum_{\tau=1}^{T}\alpha_\tau^2\mathbb{E}\|g_\tau\|^2 + \frac{1}{2\theta}\sum_{\tau=1}^{T}\alpha_\tau^2\mathbb{E}\|\mathbb{E}g_\tau - \nabla f(x_\tau)\|^2 + \frac{\theta}{2}\sum_{\tau=1}^{T}\mathbb{E}\|w_\tau - x^*\|^2\,,$$

where the last inequality follows from the last three terms being monotonically increasing with $t$. Lemma 3 of Dahan & Levy (2024) guarantees that for sequences $\{w_t\}_{t\geq 1}$ and $\{x\}_{t\geq 1}$ as defined in Equations (15) and (16), the following holds:

$$\sum_{\tau=1}^{T}\mathbb{E}\|w_t - x^*\|^2 \leq 2T\|w_1 - x^*\|^2 + 2T\eta^2\sum_{\tau=1}^{T}\alpha_\tau^2\mathbb{E}\|g_\tau\|^2 + 16\eta^2 T^2\sum_{\tau=1}^{T}\alpha_\tau^2\mathbb{E}\|\mathbb{E}g_\tau - \nabla f(x_\tau)\|^2\,,$$

which in turn implies that for $\theta = \frac{1}{4\eta T}$:

$$\alpha_{1:t}\mathbb{E}\left[f(x_t) - f^*\right] \leq \left(\frac{1}{2\eta} + \theta T\right)\|w_1 - x^*\|^2 + \left(\frac{\eta}{2} + \theta T\eta^2\right)\sum_{t=1}^{T}\alpha_t^2\mathbb{E}\|g_t\|^2$$

$$+ \left(\frac{1}{2\theta} + 8\theta\eta^2 T^2\right)\sum_{\tau=1}^{T}\alpha_\tau^2\mathbb{E}\|\mathbb{E}g_\tau - \nabla f(x_\tau)\|^2$$

$$= \frac{3\|w_1 - x^*\|^2}{4\eta} + \frac{3\eta}{4}\sum_{t=1}^{T}\alpha_t^2\mathbb{E}\|g_t\|^2 + 4\eta T\sum_{\tau=1}^{T}\alpha_\tau^2\mathbb{E}\|\mathbb{E}g_\tau - \nabla f(x_\tau)\|^2$$

$$\leq \frac{\|w_1 - x^*\|^2}{\eta} + \eta\sum_{t=1}^{T}\alpha_t^2\mathbb{E}\|g_t\|^2 + 4\eta T\sum_{\tau=1}^{T}\alpha_\tau^2\mathbb{E}\|\mathbb{E}g_\tau - \nabla f(x_\tau)\|^2\,,$$

thus concluding the proof. $\square$

# B. Proof of Theorem 4.1

In this section, we prove our main result. To simplify the presentation and analysis, we first introduce some notations:

$$X_t := \begin{pmatrix} x_t^1 & x_t^2 & \cdots & x_t^M \end{pmatrix} \in \mathbb{R}^{d\times M}, \qquad \bar{X}_t := X_t\frac{1}{M}\mathbf{1}\mathbf{1}^\top = \begin{pmatrix} \bar{x}_t & \bar{x}_t & \cdots & \bar{x}_t \end{pmatrix} \in \mathbb{R}^{d\times M},$$

$$W_t := \begin{pmatrix} w_t^1 & w_t^2 & \cdots & w_t^M \end{pmatrix} \in \mathbb{R}^{d\times M}, \quad \bar{W}_t := W_t\frac{1}{M}\mathbf{1}\mathbf{1}^\top = \begin{pmatrix} \bar{w}_t & \bar{w}_t & \cdots & \bar{w}_t \end{pmatrix} \in \mathbb{R}^{d\times M},$$

$$G_t := \begin{pmatrix} g_t^1 & g_t^2 & \cdots & g_t^M \end{pmatrix} \in \mathbb{R}^{d\times M}, \qquad \bar{G}_t := G_t\frac{1}{M}\mathbf{1}\mathbf{1}^\top = \begin{pmatrix} \bar{g}_t & \bar{g}_t & \cdots & \bar{g}_t \end{pmatrix} \in \mathbb{R}^{d\times M}\,,$$

with $\bar{x}_t = \frac{1}{M}\sum_{i=1}^{M}x_t^i$, $\bar{w}_t = \frac{1}{M}\sum_{i=1}^{M}w_t^i$, and $\bar{g}_t = \frac{1}{M}\sum_{i=1}^{M}g_t^i$.

Denoting $\delta_t = \frac{\alpha_t}{\alpha_{1:t}}$, our Decentralized Anytime SGD algorithm can be expressed using matrix notation as follows:

Local update and averaging: $\begin{cases} W_{t+\frac{1}{2}} = W_t - \eta\alpha_t G_t \\ X_{t+\frac{1}{2}} = (1-\delta_t)X_t + \delta_t W_{t+\frac{1}{2}} \end{cases}$

Gossip communication: $\begin{cases} W_{t+1} = W_{t+\frac{1}{2}}P \\ X_{t+1} = X_{t+\frac{1}{2}}P\,. \end{cases}$

We also define the expected average distance between local elements and their mean, commonly referred to as the consensus distance. Specifically, we define consensus distances for the iterates, the query points (i.e., the $\{\alpha_t\}_{t\geq 1}$-weighted iterates),

and the gradients:

$$\Gamma_t := \frac{1}{M} \sum_{i=1}^{M} \mathbb{E}\|x_t^i - \bar{x}_t\|^2 = \frac{1}{M}\mathbb{E}\|X_t - \bar{X}_t\|_F^2,$$

$$\Xi_t := \frac{1}{M} \sum_{i=1}^{M} \mathbb{E}\|w_t^i - \bar{w}_t\|^2 = \frac{1}{M}\mathbb{E}\|W_t - \bar{W}_t\|_F^2,$$

$$\Psi_t := \frac{1}{M} \sum_{i=1}^{M} \mathbb{E}\|g_t^i - \bar{g}_t\|^2 = \frac{1}{M}\mathbb{E}\|G_t - \bar{G}_t\|_F^2.$$

First, we establish that the sequence of query points average over machines $\{\bar{x}_t\}_{t\geq 1}$ is indeed an $\{\alpha_t\}_{t\geq 1}$-weighted average of the sequence of iterates average over machines $\{\bar{w}_t\}_{t\geq 1}$. This allows us to apply the results from Appendix A on Anytime SGD to analyze the consensus iterates.

**Lemma B.1.** *The sequence $\{\bar{x}_t\}_{t\geq 1}$ is an $\{\alpha_t\}_{t\geq 1}$-weighted average of the sequence $\{\bar{w}_t\}_{t\geq 1}$.*

*Proof.* Using matrix notation and the linearity of averaging, we have:

$$\bar{X}_{t+1} = \bar{X}_{t+\frac{1}{2}}P = \bar{X}_{t+\frac{1}{2}} = \frac{\alpha_{1:t-1}}{\alpha_{1:t}}\bar{X}_t + \frac{\alpha_t}{\alpha_{1:t}}\bar{W}_{t+\frac{1}{2}} = \frac{\alpha_{1:t-1}}{\alpha_{1:t}}\bar{X}_t + \frac{\alpha_t}{\alpha_{1:t}}\bar{W}_{t+\frac{1}{2}}P = \frac{\alpha_{1:t-1}}{\alpha_{1:t}}\bar{X}_t + \frac{\alpha_t}{\alpha_{1:t}}\bar{W}_{t+1},$$

where the second and fourth equalities follow from Lemma D.1, implying that gossip communication preserves averages. $\square$

Next, we establish the convergence of Algorithm 2, as stated in Theorem 4.1, which we restate here for ease of reference.

**Theorem 4.1.** *Under Assumptions 2.1-2.3, consider Algorithm 2 with linear weights $\alpha_t = t$ and a learning rate given by*

$$\eta = \min\left\{\frac{1}{24LT}, \frac{\rho^2}{K}, \frac{D_1\sqrt{M}}{\sqrt{3}\sigma T^{3/2}}, \sqrt{\frac{D_1}{2K\tilde{\sigma}}}\frac{\rho}{T}\right\},$$

*where $D_1^2 := \|w_1 - x^*\|^2$, $K^2 := 5120L^2$, and $\tilde{\sigma}^2 := 2\sigma^2 + \zeta^2$. Then, for all $T \geq 1$, the following bound holds:*

$$\mathbb{E}[f(\bar{x}_T) - f^*] \leq \mathcal{O}\left(\frac{\sigma D_1}{\sqrt{MT}} + \frac{D_1^{3/2}\sqrt{L\tilde{\sigma}}}{\rho T} + \frac{LD_1^2}{T}\right),$$

*where $\bar{x}_T := \frac{1}{M}\sum_{i=1}^{M} x_T^i$.*

*Proof.* For any $t \in [T]$, define $\Delta_t := \mathbb{E}[f(\bar{x}_t) - f^*]$. We analyze the consensus iterates, which (according to Lemma B.1) follow the structure of Anytime SGD as defined in Equations (15) and (16), namely,

$$\bar{w}_{t+1} = \bar{w}_t - \eta\alpha_t\bar{g}_t, \quad \text{and} \quad \bar{x}_{t+1} = (1-\delta_t)\bar{x}_t + \delta_t\bar{w}_{t+1}.$$

Therefore, by Lemma A.3, we have:

$$\alpha_{1:t}\Delta_t \leq \frac{\|\bar{w}_1 - x^*\|^2}{\eta} + \eta\sum_{\tau=1}^{T}\alpha_\tau^2\mathbb{E}\|\bar{g}_\tau\|^2 + 4\eta T\sum_{\tau=1}^{T}\alpha_\tau^2\mathbb{E}\|\mathbb{E}\bar{g}_\tau - \nabla f(\bar{x}_\tau)\|^2$$

$$= \frac{D_1^2}{\eta} + \eta\underbrace{\sum_{\tau=1}^{T}\alpha_\tau^2\mathbb{E}\|\bar{g}_\tau\|^2}_{(A)} + 4\eta T \cdot B_T, \tag{19}$$

where $B_T := \sum_{\tau=1}^{T}\alpha_\tau^2\mathbb{E}\|\mathbb{E}\bar{g}_\tau - \nabla f(\bar{x}_\tau)\|^2$, and we also used $\bar{w}_1 = w_1$.

**Bounding** $(A)$. We focus on $\mathbb{E}\|\bar{g}_\tau\|^2$. By adding and subtracting $\mathbb{E}\bar{g}_\tau$ and $\nabla f(\bar{x}_\tau)$, and applying the inequality $\|a + b + c\|^2 \le 3\|a\|^2 + 3\|b\|^2 + 3\|c\|^2$, we obtain:

$$
\begin{aligned}
\mathbb{E}\|\bar{g}_\tau\|^2 &= \mathbb{E}\|\bar{g}_\tau - \mathbb{E}\bar{g}_\tau + \mathbb{E}\bar{g}_\tau - \nabla f(\bar{x}_\tau) + \nabla f(\bar{x}_\tau)\|^2 \\
&\le 3\mathbb{E}\|\bar{g}_\tau - \mathbb{E}\bar{g}_\tau\|^2 + 3\mathbb{E}\|\mathbb{E}\bar{g}_\tau - \nabla f(\bar{x}_\tau)\|^2 + 3\mathbb{E}\|\nabla f(\bar{x}_\tau)\|^2 \\
&\le \frac{3\sigma^2}{M} + 3\mathbb{E}\|\mathbb{E}\bar{g}_\tau - \nabla f(\bar{x}_\tau)\|^2 + 3\mathbb{E}\|\nabla f(\bar{x}_\tau)\|^2 \ ,
\end{aligned}
$$

where the last inequality holds because $\left\{g_\tau^i - \nabla f_i(x_\tau^i)\right\}_{i \in [M]}$ are independent, zero-mean, and have variance at most $\sigma^2$:

$$
\mathbb{E}\|\bar{g}_\tau - \mathbb{E}\bar{g}_\tau\|^2 = \mathbb{E}\left\|\frac{1}{M}\sum_{i=1}^M \left(g_\tau^i - \nabla f_i(x_\tau^i)\right)\right\|^2 = \frac{1}{M^2}\sum_{i=1}^M \mathbb{E}\left\|g_t^i - \nabla f_i(x_\tau^i)\right\|^2 \le \frac{\sigma^2}{M} \ .
$$

Thus, $(A)$ is bounded as follows,

$$
\sum_{\tau=1}^T \alpha_t^2 \mathbb{E}\|\bar{g}_\tau\|^2 \le \frac{3\sigma^2}{M}\sum_{\tau=1}^T \alpha_\tau^2 + 3B_T + 3\sum_{\tau=1}^T \alpha_\tau^2 \mathbb{E}\|\nabla f(\bar{x}_\tau)\|^2 \ .
$$

Plugging this bound back into (19), we get:

$$
\begin{aligned}
\alpha_{1:t}\Delta_t &\le \frac{D_1^2}{\eta} + \eta\left(\frac{3\sigma^2}{M}\sum_{\tau=1}^T \alpha_\tau^2 + 3V_T + 3\sum_{\tau=1}^T \alpha_\tau^2 \mathbb{E}\|\nabla f(\bar{x}_\tau)\|^2\right) + 4\eta T \cdot V_T \\
&= \frac{D_1^2}{\eta} + \frac{3\sigma^2 \eta}{M}\sum_{\tau=1}^T \alpha_\tau^2 + (3\eta + 4\eta T) \cdot B_T + 3\eta\sum_{\tau=1}^T \alpha_\tau^2 \mathbb{E}\|\nabla f(\bar{x}_\tau)\|^2 \\
&\le \frac{D_1^2}{\eta} + \frac{3\sigma^2 \eta}{M}\sum_{\tau=1}^T \alpha_\tau^2 + 8\eta T \cdot B_T + 3\eta\sum_{\tau=1}^T \alpha_\tau^2 \mathbb{E}\|\nabla f(\bar{x}_\tau)\|^2 \ . \quad (20)
\end{aligned}
$$

For linear weights $\alpha_t = t$ and learning rate upper bounded by $\frac{\rho^2}{K}$, Lemma 4.3 provides the following bound on $B_T$ (see proof in Appendix B.2 below):

$$
B_T \le \frac{K^2 \tilde{\sigma}^2 \eta^2}{2\rho^4}\sum_{\tau=1}^T \alpha_\tau^2 \ .
$$

Substituting this bound back to Equation (20), we obtain:

$$
\alpha_{1:t}\Delta_t \le \frac{D_1^2}{\eta} + \frac{3\sigma^2 \eta}{M}\sum_{\tau=1}^T \alpha_\tau^2 + \frac{4K^2 \tilde{\sigma}^2 \eta^3 T}{\rho^4}\sum_{\tau=1}^T \alpha_\tau^2 + 3\eta\sum_{\tau=1}^T \alpha_\tau^2 \mathbb{E}\|\nabla f(\bar{x}_\tau)\|^2 \ .
$$

The rightmost term is be bounded as follows:

$$
\sum_{\tau=1}^T \alpha_t^2 \mathbb{E}\|\nabla f(\bar{x}_\tau)\|^2 \le 2L\sum_{\tau=1}^T \alpha_\tau^2 \Delta_\tau \le 4L\sum_{\tau=1}^T \alpha_{1:\tau}\Delta_\tau \ ,
$$

where the first inequality follows from Lemma D.2 (i.e., $\|\nabla f(\bar{x}_\tau)\|^2 \le 2L(f(\bar{x}_\tau) - f^*)$), and the second inequality arises from $\alpha_t^2 = t^2 \le t(t+1) = 2\alpha_{1:t}$. Therefore,

$$
\begin{aligned}
\alpha_{1:t}\Delta_t &\le \frac{D_1^2}{\eta} + \frac{3\sigma^2 \eta}{M}\sum_{\tau=1}^T \alpha_\tau^2 + \frac{4K^2 \tilde{\sigma}^2 \eta^3 T}{\rho^4}\sum_{\tau=1}^T \alpha_\tau^2 + 12L\eta\sum_{\tau=1}^T \alpha_{1:\tau}\Delta_\tau \\
&\le \frac{D_1^2}{\eta} + \frac{3\sigma^2 \eta}{M}\sum_{\tau=1}^T \alpha_\tau^2 + \frac{4K^2 \tilde{\sigma}^2 \eta^3 T}{\rho^4}\sum_{\tau=1}^T \alpha_\tau^2 + \frac{1}{2T}\sum_{\tau=1}^T \alpha_{1:\tau}\Delta_\tau \ ,
\end{aligned}
$$

where the last inequality holds true for $\eta \leq \frac{1}{24LT}$. Applying Lemma D.3 with $a_t = \alpha_{1:t}\Delta_t$ and $b = \frac{D_1^2}{\eta} + \frac{3\sigma^2\eta}{M}\sum_{\tau=1}^{T}\alpha_\tau^2 + \frac{4K^2\tilde{\sigma}^2\eta^3 T}{\rho^4}\sum_{\tau=1}^{T}\alpha_\tau^2$ yields (for $t = T$):

$$\alpha_{1:T}\Delta_T \leq 2b$$

$$\leq \frac{2D_1^2}{\eta} + \frac{6\sigma^2\eta}{M}\sum_{\tau=1}^{T}\alpha_\tau^2 + \frac{8K^2\tilde{\sigma}^2\eta^3 T}{\rho^4}\sum_{\tau=1}^{T}\alpha_\tau^2$$

$$\leq \frac{2D_1^2}{\eta} + \frac{6\sigma^2\eta T^3}{M} + \frac{8K^2\tilde{\sigma}^2\eta^3 T^4}{\rho^4} \,,$$

where the second inequality holds as $\sum_{t=1}^{T}\alpha_t^2 = \sum_{t=1}^{T}t^2 \leq T^3$. Thus, employing Lemma D.6 with $A = 2D_1^2$, $B = \frac{6\sigma^2 T^3}{M}$, $C = \frac{8K^2\tilde{\sigma}^2 T^4}{\rho^4}$, and $\eta_1 = \min\left\{\frac{1}{24LT}, \frac{\rho^2}{K}\right\}$, we get:

$$\alpha_{1:T}\Delta_T \leq \frac{A}{\eta_1} + 2\sqrt{AB} + 2A^{3/4}C^{1/4}$$

$$\leq 2D_1^2\left(24LT + \frac{K}{\rho^2}\right) + 2\sqrt{\frac{12D_1^2\sigma^2 T^3}{M}} + 2\left(2D_1^2\right)^{3/4}\left(\frac{8K^2\tilde{\sigma}^2 T^4}{\rho^4}\right)^{1/4}$$

$$= \frac{4\sqrt{3}D_1\sigma T^{3/2}}{\sqrt{M}} + \frac{4\sqrt{2}D_1^{3/2}\sqrt{K\tilde{\sigma}}T}{\rho} + 48LD_1^2 T + \frac{2KD_1^2}{\rho^2} \,.$$

Finally, dividing by $\alpha_{1:T} = \frac{T(T+1)}{2} \geq \frac{T^2}{2}$, we obtain the result as:

$$\Delta_T = \mathbb{E}[f(\bar{x}_T) - f^*] \leq \frac{8\sqrt{3}D_1\sigma}{\sqrt{MT}} + \frac{8\sqrt{2}D_1^{3/2}\sqrt{K\tilde{\sigma}}}{\rho T} + \frac{96LD_1^2}{T} + \frac{4KD_1^2}{\rho^2 T^2}$$

$$= \mathcal{O}\left(\frac{D_1\sigma}{\sqrt{MT}} + \frac{D_1^{3/2}\sqrt{L(\sigma+\zeta)}}{\rho T} + \frac{LD_1^2}{T} + \frac{LD_1^2}{\rho^2 T^2}\right) \,.$$

$\square$

## B.1. Proof of Corollary 4.2

Next, we demonstrate the convergence of the local iterates.

*Proof.* Let $i \in [M]$. By the smoothness of the $f$, it holds that:

$$\mathbb{E}[f(x_T^i) - f^*] = \mathbb{E}[f(x_T^i) - f(\bar{x}_T)] + \mathbb{E}[f(\bar{x}_T) - f^*]$$

$$\leq \mathbb{E}\left[\nabla f(\bar{x}_T)^\top(x_T^i - \bar{x}_T) + \frac{L}{2}\left\|x_T^i - \bar{x}_T\right\|^2\right] + \mathbb{E}[f(\bar{x}_T) - f^*]$$

$$\leq \frac{1}{2\theta}\mathbb{E}\left\|\nabla f(\bar{x}_T)\right\|^2 + \frac{\theta+L}{2}\mathbb{E}\left\|x_T^i - \bar{x}_T\right\|^2 + \mathbb{E}[f(\bar{x}_T) - f^*] \,,$$

where the last inequality holds for any $\theta > 0$ due to Young's inequality, $a^\top b \leq \frac{1}{2\theta}\|a\|^2 + \frac{\theta}{2}\|b\|^2$. Setting $\theta = L$ and using Lemma D.2 to upper bound $\|\nabla f(\bar{x}_T)\|^2$ by $2L(f(\bar{x}_T) - f^*)$, we get:

$$\mathbb{E}[f(x_T^i) - f^*] \leq \frac{1}{2L}\mathbb{E}\left\|\nabla f(\bar{x}_T)\right\|^2 + L\mathbb{E}\left\|x_T^i - \bar{x}_T\right\|^2 + \mathbb{E}[f(\bar{x}_T) - f^*]$$

$$\leq 2\mathbb{E}[f(\bar{x}_T) - f^*] + L\mathbb{E}\left\|x_T^i - \bar{x}_T\right\|^2$$

$$\leq 2\mathbb{E}[f(\bar{x}_T) - f^*] + L\sum_{i=1}^{M}\mathbb{E}\left\|x_T^i - \bar{x}_T\right\|^2$$

$$= 2\mathbb{E}[f(\bar{x}_T) - f^*] + LM\Gamma_T \,.$$

Using Lemma C.3, which applies under the conditions of Theorem 4.1, we can bound $\Gamma_T$ by $2560\tilde{\sigma}^2\eta^2/\rho^4$, where $\tilde{\sigma}^2 := 2\sigma^2 + \zeta^2$. In addition, the learning rate in Theorem 4.1 satisfies $\eta^2 \leq D_1\rho^2/2K\tilde{\sigma}T^2$, where $K := \sqrt{5120}L$. Therefore, we obtain:

$$\mathbb{E}[f(x_T^i) - f^*] \leq 2\mathbb{E}[f(\bar{x}_T) - f^*] + \frac{2560LM\tilde{\sigma}^2\eta^2}{\rho^4} \leq 2\mathbb{E}[f(\bar{x}_T) - f^*] + \frac{8\sqrt{5}MD_1\tilde{\sigma}}{\rho^2T^2} .$$

Plugging the bound on $\mathbb{E}[f(\bar{x}_T) - f^*]$ from Theorem 4.1 establishes the result:

$$\mathbb{E}[f(x_T^i) - f^*] \leq \frac{16\sqrt{3}D_1\sigma}{\sqrt{MT}} + \frac{16\sqrt{2}D_1^{3/2}\sqrt{K\tilde{\sigma}}}{\rho T} + \frac{192LD_1^2}{T} + \frac{8KD_1^2}{\rho^2T^2} + \frac{8\sqrt{5}MD_1\tilde{\sigma}}{\rho^2T^2}$$

$$= \mathcal{O}\left(\frac{D_1\sigma}{\sqrt{MT}} + \frac{D_1^{3/2}\sqrt{L\tilde{\sigma}}}{\rho T} + \frac{LD_1^2}{T} + \frac{MD_1\tilde{\sigma}}{\rho^2T^2}\right) .$$

$\square$

In terms on the total of samples $N = MT$, the established rate is given by:

$$\mathbb{E}[f(x_T^i) - f^*] \leq \mathcal{O}\left(\frac{D_1\sigma}{\sqrt{N}} + \frac{MD_1^{3/2}\sqrt{L\tilde{\sigma}}}{\rho N} + \frac{MLD_1^2}{N} + \frac{M^3D_1\tilde{\sigma}}{\rho^2N^2}\right) .$$

Therefore, ignoring the dependence on $L, D_1, \sigma$ and $\zeta$ (i.e., assuming they all equal 1), the parallelism bound (i.e., the maximal asymptotic $M$ for which the rate is $\mathcal{O}(1/\sqrt{N})$) is:

$$M \leq \mathcal{O}\left(\min\left\{\rho\sqrt{N}, \sqrt{N}, \rho^{2/3}\sqrt{N}\right\}\right) = \mathcal{O}(\rho\sqrt{N}) .$$

## B.2. Proof of Lemma 4.3

*Proof.* We aim to prove that:

$$B_T := \mathbb{E}\left[\sum_{\tau=1}^{T}\alpha_\tau^2\|\mathbb{E}\bar{g}_\tau - \nabla f(\bar{x}_\tau)\|^2\right] \leq \frac{K^2\tilde{\sigma}^2\eta^2}{2\rho^4}\sum_{\tau=1}^{T}\alpha_\tau^2 ,$$

where $K^2 := 5120L^2$ and $\tilde{\sigma}^2 := 2\sigma^2 + \zeta^2$.

Focusing on $\mathbb{E}\|\mathbb{E}\bar{g}_\tau - \nabla f(\bar{x}_\tau)\|^2$ and using the convexity of $\|\cdot\|^2$, we apply Jensen's inequality to obtain:

$$\mathbb{E}\|\mathbb{E}\bar{g}_\tau - \nabla f(\bar{x}_\tau)\|^2 = \mathbb{E}\left\|\frac{1}{M}\sum_{i=1}^{M}\left(\nabla f_i(x_\tau^i) - \nabla f_i(\bar{x}_\tau)\right)\right\|^2$$

$$\leq \frac{1}{M}\sum_{i=1}^{M}\mathbb{E}\|\nabla f_i(x_\tau^i) - \nabla f_i(\bar{x}_\tau)\|^2$$

$$\leq \frac{L^2}{M}\sum_{i=1}^{M}\mathbb{E}\|x_\tau^i - \bar{x}_\tau\|^2$$

$$= L^2\Gamma_\tau ,$$

where the second inequality follows from the smoothness of $f_i$. Using Lemma C.3, which applies specifically to linear weights $\alpha_t = t$ and a learning rate bounded as $\eta \leq \frac{\rho^2}{K} = \frac{\rho^2}{8\sqrt{80}L}$ we bound $\Gamma_\tau$ and derive a corresponding bound for $B_T$:

$$B_T = \sum_{\tau=1}^{T}\alpha_\tau^2\mathbb{E}\|\mathbb{E}\bar{g}_\tau - \nabla f(\bar{x}_\tau)\|^2 \leq L^2\sum_{\tau=1}^{T}\alpha_\tau^2\Gamma_\tau \leq \frac{2560L^2\tilde{\sigma}^2\eta^2}{\rho^4}\sum_{\tau=1}^{T}\alpha_\tau^2 = \frac{K^2\tilde{\sigma}^2\eta^2}{2\rho^4}\sum_{\tau=1}^{T}\alpha_\tau^2 ,$$

which concludes the proof. $\square$

## C. Consensus Recursion Analysis

In this section, we analyze the consensus distances $\Gamma_t, \Xi_t$, and $\Psi_t$. We begin by observing that $\Gamma_t$ follows a recursive relation (Equation (24)), where $\Gamma_{t+1}$ depends not only on $\Gamma_t$ but also on $\Xi_t$ and $\Psi_t$. Similarly, $\Xi_t$ satisfies its own recursion, with $\Xi_{t+1}$ depending on $\Xi_t$ and $\Psi_t$ (Lemma C.4).

To simplify the analysis, we first solve the recursion for $\Xi_t$, allowing us to express $\Gamma_t$ in terms of $\Psi_t$ alone, eliminating the dependence on $\Xi_t$ (Lemma C.1). Finally, in Lemma C.2, we bound $\Psi_t$ in terms of the problem parameters ($\sigma$ and $\zeta$) and $\Gamma_t$, enabling us to explicitly solve the recursion for $\Gamma_t$ (Lemma C.3).

### C.1. Consensus of the Query Points

The next result establishes a recursion for the consensus distance of the query points $X_t$.

**Lemma C.1.** *For all $t \geq 1$, the consensus distance of the query points (i.e., averaged iterates) $\Gamma_t$ satisfies the following recursion:*

$$\Gamma_{t+1} \leq \left(1 - \frac{\rho}{2}\right)\Gamma_t + \frac{4\eta^2\alpha_t^2\delta_t^2}{\rho}\left(\Psi_t + \frac{2}{\rho}\sum_{\tau=1}^{t}\left(1 - \frac{\rho}{2}\right)^{t-1-\tau}\Psi_\tau\right).$$

*Proof.* The gossip averaging of the query points leads to the following inequality:

$$M\Gamma_{t+1} = \mathbb{E}\|X_{t+1} - \bar{X}_{t+1}\|_F^2 = \mathbb{E}\|X_{t+\frac{1}{2}}P - \bar{X}_{t+\frac{1}{2}}\|_F^2 \leq (1-\rho)\mathbb{E}\|X_{t+\frac{1}{2}} - \bar{X}_{t+\frac{1}{2}}\|_F^2, \tag{21}$$

where we used Lemma D.1, stating that $\bar{X}_{t+\frac{1}{2}}P = \bar{X}_{t+\frac{1}{2}}$, and the mixing property of the gossip matrix (Property 2.6). Substituting the query points averaging, $X_{t+\frac{1}{2}} = (1-\delta_t)X_t + \delta_t W_{t+\frac{1}{2}}$, and using $\|a + b\|^2 \leq (1+\beta^{-1})\|a\|^2 + (1+\beta)\|b\|^2$ (which holds for any $\beta > 0$), we obtain:

$$
\begin{aligned}
M\Gamma_{t+1} &\leq (1-\rho)\mathbb{E}\|X_{t+\frac{1}{2}} - \bar{X}_{t+\frac{1}{2}}\|_F^2 \\
&= (1-\rho)\mathbb{E}\|(1-\delta_t)X_t + \delta_t W_{t+\frac{1}{2}} - ((1-\delta_t)\bar{X}_t + \delta_t\bar{W}_{t+\frac{1}{2}})\|_F^2 \\
&\leq (1-\rho)\left(1 + \frac{1}{\beta}\right)(1-\delta_t)^2\mathbb{E}\|X_t - \bar{X}_t\|_F^2 + (1-\rho)(1+\beta)\delta_t^2\mathbb{E}\|W_{t+\frac{1}{2}} - \bar{W}_{t+\frac{1}{2}}\|_F^2 \\
&\leq (1-\rho)\left(1 + \frac{1}{\beta}\right)M\Gamma_t + (1-\rho)(1+\beta)\delta_t^2\underbrace{\mathbb{E}\|W_{t+\frac{1}{2}} - \bar{W}_{t+\frac{1}{2}}\|_F^2}_{(\star)},
\end{aligned}
\tag{22}
$$

where the last inequality follows from $(1 - \delta_t)^2 \leq 1$. Focusing on $(\star)$ and plugging the iterate update rule, we have for all $\gamma > 0$:

$$
\begin{aligned}
\mathbb{E}\|W_{t+\frac{1}{2}} - \bar{W}_{t+\frac{1}{2}}\|_F^2 &= \mathbb{E}\|W_t - \eta\alpha_t G_t - (\bar{W}_t - \eta\alpha_t\bar{G}_t)\|_F^2 \\
&\leq \left(1 + \frac{1}{\gamma}\right)\mathbb{E}\|W_t - \bar{W}_t\|_F^2 + (1+\gamma)\eta^2\alpha_t^2\mathbb{E}\|G_t - \bar{G}_t\|_F^2 \\
&= \left(1 + \frac{1}{\gamma}\right)M\Xi_t + (1+\gamma)\eta^2\alpha_t^2 M\Psi_t.
\end{aligned}
\tag{23}
$$

Substituting (23) into (22), setting $\beta = 2/\rho$ and $\gamma = 1$, and dividing by $M$, we get:

$$\Gamma_{t+1} \leq (1-\rho)\left(1 + \frac{\rho}{2}\right)\Gamma_t + (1-\rho)\left(1 + \frac{2}{\rho}\right)\delta_t^2\left(2\Xi_t + 2\eta^2\alpha_t^2\Psi_t\right) \leq \left(1 - \frac{\rho}{2}\right)\Gamma_t + \frac{4\delta_t^2}{\rho}\left(\Xi_t + \eta^2\alpha_t^2\Psi_t\right), \tag{24}$$

where in the second inequality we used $(1 - \rho)(1 + \frac{\rho}{2}) = 1 - \frac{\rho}{2} - \frac{\rho^2}{2} \leq 1 - \frac{\rho}{2}$ and $(1 - \rho)(1 + \frac{2}{\rho}) = -1 - \rho + \frac{2}{\rho} \leq \frac{2}{\rho}$. Note that we derived a recursion for $\Gamma_t$, which also involves $\Xi_t$. The consensus distance of the iterates, $\Xi_t$, satisfies its own

recursion as stated in Lemma C.4, from which an explicit bound is provided in Lemma C.5. Substituting this bound yields:

$$
\begin{aligned}
\Gamma_{t+1} &\leq \left(1 - \frac{\rho}{2}\right)\Gamma_t + \frac{4\delta_t^2}{\rho}\left(\frac{2\eta^2\alpha_t^2}{\rho}\sum_{\tau=1}^{t-1}\left(1 - \frac{\rho}{2}\right)^{t-1-\tau}\Psi_\tau + \eta^2\alpha_t^2\Psi_t\right) \\
&= \left(1 - \frac{\rho}{2}\right)\Gamma_t + \frac{4\eta^2\alpha_t^2\delta_t^2}{\rho}\left(\Psi_t + \frac{2}{\rho}\sum_{\tau=1}^{t-1}\left(1 - \frac{\rho}{2}\right)^{t-1-\tau}\Psi_\tau\right),
\end{aligned}
\tag{25}
$$

thus concluding the proof. □

Note that the sequence $\Psi_1, \ldots, \Psi_t$ appears in the recursion given in Lemma C.1. We next provide an upper bound on $\Psi_t$ in terms of $\Gamma_t$, which will enable us to derive an explicit bound for $\Gamma_t$.

**Lemma C.2.** *For every $t \geq 1$, $\Psi_t \leq 5\left(2\sigma^2 + \zeta^2\right) + 10L^2\Gamma_t$.*

*Proof.* Using $\|\sum_{n=1}^N u_n\|^2 \leq N\sum_{n=1}^N \|u_n\|^2$, we have:

$$
\begin{aligned}
\Psi_t &= \frac{1}{M}\sum_{i=1}^M \mathbb{E}\|g_t^i - \bar{g}_t\|^2 \\
&= \frac{1}{M}\sum_{i=1}^M \mathbb{E}\|g_t^i - \mathbb{E}g_t^i + \mathbb{E}g_t^i - \bar{g}_t + \mathbb{E}\bar{g}_t - \mathbb{E}\bar{g}_t - \nabla f_i(\bar{x}_t) + \nabla f_i(\bar{x}_t) - \nabla f(\bar{x}_t) + \nabla f(\bar{x}_t)\|^2 \\
&\leq \frac{5}{M}\sum_{i=1}^M \left(\mathbb{E}\|g_t^i - \mathbb{E}g_t^i\|^2 + \mathbb{E}\|\bar{g}_t - \mathbb{E}\bar{g}_t\|^2 + \mathbb{E}\|\mathbb{E}g_t^i - \nabla f_i(\bar{x}_t)\|^2 + \mathbb{E}\|\mathbb{E}\bar{g}_t - \nabla f(\bar{x}_t)\|^2 + \mathbb{E}\|\nabla f_i(\bar{x}_t) - \nabla f(\bar{x}_t)\|^2\right).
\end{aligned}
\tag{26}
$$

Next, we individually bound each sum.

**Bounding $\frac{1}{M}\sum_{i=1}^M \mathbb{E}\|g_t^i - \mathbb{E}g_t^i\|^2$.** By the bounded variance assumption, it holds that:

$$
\frac{1}{M}\sum_{i=1}^M \mathbb{E}\|g_t^i - \mathbb{E}g_t^i\|^2 = \frac{1}{M}\sum_{i=1}^M \mathbb{E}\|\nabla f_i(x_t^i, z_t^i) - \nabla f_i(x_t^i)\|^2 \leq \frac{1}{M}\sum_{i=1}^M \sigma^2 = \sigma^2.
$$

**Bounding $\mathbb{E}\|\bar{g}_t - \mathbb{E}\bar{g}_t\|^2$.** Since $g_t^i - \mathbb{E}g_t^i$ are independent, zero mean, and have variance bounded by $\sigma^2$, it follows that:

$$
\mathbb{E}\|\bar{g}_t - \mathbb{E}\bar{g}_t\|^2 = \mathbb{E}\left\|\frac{1}{M}\sum_{i=1}^M\left(g_t^i - \mathbb{E}g_t^i\right)\right\|^2 = \frac{1}{M^2}\sum_{i=1}^M \mathbb{E}\|g_t^i - \mathbb{E}g_t^i\|^2 \leq \frac{\sigma^2}{M}.
$$

**Bounding $\frac{1}{M}\sum_{i=1}^M \mathbb{E}\|\mathbb{E}g_t^i - \nabla f_i(\bar{x}_t)\|^2$.** By the smoothness of $f_i$,

$$
\frac{1}{M}\sum_{i=1}^M \mathbb{E}\|\mathbb{E}g_t^i - \nabla f_i(\bar{x}_t)\|^2 = \frac{1}{M}\sum_{i=1}^M \mathbb{E}\|\nabla f_i(x_t^i) - \nabla f_i(\bar{x}_t)\|^2 \leq \frac{L^2}{M}\sum_{i=1}^M \mathbb{E}\|x_t^i - \bar{x}_t\|^2 = L^2\Gamma_t.
$$

**Bounding $\mathbb{E}\|\mathbb{E}\bar{g}_t - \nabla f(\bar{x}_t)\|^2$.** By Jensen's inequality and the smoothness of each $f_i$, we have:

$$
\mathbb{E}\|\mathbb{E}\bar{g}_t - \nabla f(\bar{x}_t)\|^2 = \mathbb{E}\left\|\frac{1}{M}\sum_{i=1}^M\left(\nabla f_i(x_t^i) - \nabla f_i(\bar{x}_t)\right)\right\|^2 \leq \frac{1}{M}\sum_{i=1}^M \mathbb{E}\|\nabla f_i(x_t^i) - \nabla f_i(\bar{x}_t)\|^2 \leq L^2\Gamma_t.
$$

**Bounding** $\frac{1}{M} \sum_{i=1}^{M} \mathbb{E}\|\nabla f_i(\bar{x}_t) - \nabla f(\bar{x}_t)\|^2$**.** By the heterogeneity assumption (Assumption 2.3),

$$\frac{1}{M} \sum_{i=1}^{M} \mathbb{E}\|\nabla f_i(\bar{x}_t) - \nabla f(\bar{x}_t)\|^2 \leq \zeta^2 .$$

Substituting these bounds back into Equation (26) implies the result as,

$$\Psi_t \leq 5\left(\sigma^2 + \frac{\sigma^2}{M} + 2L^2\Gamma_t + \zeta^2\right) \leq 5\left(2\sigma^2 + \zeta^2\right) + 10L^2\Gamma_t .$$

$\square$

Using this, we now derive an explicit bound for $\Gamma_t$, specifically for linear weights and a sufficiently small learning rate.

**Lemma C.3.** *For linear weights $\{\alpha_t = t\}_{t \geq 1}$, and a learning rate $\eta \leq \frac{\rho^2}{8\sqrt{80}L}$, the consensus distance of the query points $\Gamma_t$ is bounded as:*

$$\Gamma_t \leq \frac{2560\tilde{\sigma}^2\eta^2}{\rho^4} ,$$

*where $\tilde{\sigma}^2 := 2\sigma^2 + \zeta^2$.*

*Proof.* First, observe that for linear weights $\alpha_t = t$, we have $\alpha_{1:t} = \sum_{\tau=1}^{t} \tau = \frac{t(t+1)}{2}$, which implies that for all $t \geq 1$:

$$\delta_t = \frac{\alpha_t}{\alpha_{1:t}} = \frac{2}{t+1}, \quad \text{and} \quad \alpha_t\delta_t = \frac{2t}{t+1} \leq 2 .$$

Thus, from Lemma C.1, we get:

$$\Gamma_{t+1} \leq \left(1 - \frac{\rho}{2}\right)\Gamma_t + \frac{4\eta\alpha_t^2\delta_t^2}{\rho}\left(\Psi_t + \frac{2}{\rho}\sum_{\tau=1}^{t-1}\left(1 - \frac{\rho}{2}\right)^{t-1-\tau}\Psi_\tau\right)$$

$$\leq \left(1 - \frac{\rho}{2}\right)\Gamma_t + \frac{16\eta}{\rho}\left(\Psi_t + \frac{2}{\rho}\sum_{\tau=1}^{t-1}\left(1 - \frac{\rho}{2}\right)^{t-1-\tau}\Psi_\tau\right) .$$

where the second inequality follows from $\alpha_t^2\delta_t^2 \leq 4$. Let $\tilde{\sigma}^2 := 2\sigma^2 + \zeta^2$. Then, by Lemma C.2, for all $t \geq$, we have $\Psi_t \leq 5\tilde{\sigma}^2 + 10L^2\Gamma_t$. Substituting this bound yields:

$$\Gamma_{t+1} \leq \left(1 - \frac{\rho}{2}\right)\Gamma_t + \frac{80\eta^2}{\rho}\left(\tilde{\sigma}^2 + 2L^2\Gamma_t + \frac{2}{\rho}\sum_{\tau=1}^{t-1}\left(1 - \frac{\rho}{2}\right)^{t-1-\tau}(\tilde{\sigma}^2 + 2L^2\Gamma_\tau)\right)$$

$$= \left(1 - \frac{\rho}{2} + 80L^2\eta^2 \cdot \frac{2}{\rho}\right)\Gamma_t + 80L^2\eta^2\left(\frac{2}{\rho}\right)^2 \cdot \sum_{\tau=1}^{t-1}\left(1 - \frac{\rho}{2}\right)^{t-1-\tau}\Gamma_\tau + \frac{80\eta^2}{\rho}\left(1 + \frac{2}{\rho}\sum_{\tau=1}^{t-1}\left(1 - \frac{\rho}{2}\right)^{t-1-\tau}\right)\tilde{\sigma}^2$$

$$\leq \left(1 - \frac{\rho}{2} + c_1^2\eta^2 \cdot \frac{2}{\rho}\right)\Gamma_t + c_1^2\eta^2\left(\frac{2}{\rho}\right)^2 \cdot \sum_{\tau=1}^{t-1}\left(1 - \frac{\rho}{2}\right)^{t-1-\tau}\Gamma_\tau + \frac{c_2^2\eta^2}{\rho}\left(1 + \frac{4}{\rho^2}\right)$$

$$\leq \left(1 - \frac{\rho}{2} + c_1^2\eta^2 \cdot \frac{2}{\rho}\right)\Gamma_t + c_1^2\eta^2\left(\frac{2}{\rho}\right)^2 \cdot \sum_{\tau=1}^{t-1}\left(1 - \frac{\rho}{2}\right)^{t-1-\tau}\Gamma_\tau + c_2^2\eta^2\left(\frac{2}{\rho}\right)^3 , \tag{27}$$

where $c_1^2 := 80L^2$, $c_2^2 := 80\tilde{\sigma}^2$, the final inequality holds because $1 \leq \frac{4}{\rho^2}$, and the second inequality stems from the following bound on the geometric sum:

$$\sum_{\tau=1}^{t-1}\left(1 - \frac{\rho}{2}\right)^{t-1-\tau} = \sum_{\tau=0}^{t-2}\left(1 - \frac{\rho}{2}\right)^\tau \leq \sum_{\tau=0}^{\infty}\left(1 - \frac{\rho}{2}\right)^\tau = \frac{2}{\rho} .$$

Equation (27) exhibits a recursive structure as in Lemma D.5. Since $\eta \leq \frac{\rho^2}{8\sqrt{80}L}$, the condition required to apply the lemma is satisfied. Applying the lemma with $a_t = \Gamma_t$ and $\kappa = \frac{\rho}{2}$ yields the final result:

$$\Gamma_t \leq \frac{2c_2^2\eta^2}{\kappa^4} = \frac{2560\tilde{\sigma}^2\eta^2}{\rho^4} .$$

$\square$

### C.2. Consensus of the Iterates

The following lemma provides a recursive relation for the consensus distance of the iterates $W_t$.

**Lemma C.4.** *The consensus distance for the iterates $\Xi_t$, satisfies the following recursive relation:*

$$\Xi_{t+1} \leq \left(1 - \frac{\rho}{2}\right)\Xi_t + \frac{2\eta^2\alpha_t^2}{\rho}\Psi_t .$$

*Proof.* Similarly to Equation (21), we have by the gossip averaging of the iterates:

$$M\Xi_{t+1} = \mathbb{E}\|W_{t+1} - \bar{W}_{t+1}\|_F^2 = \mathbb{E}\|W_{t+\frac{1}{2}}P - \bar{W}_{t+\frac{1}{2}}\|_F^2 \leq (1 - \rho)\mathbb{E}\|W_{t+\frac{1}{2}} - \bar{W}_{t+\frac{1}{2}}\|_F^2 ,$$

where the second equality is due to Lemma D.1, stating that $\bar{W}_{t+\frac{1}{2}}P = \bar{W}_{t+\frac{1}{2}}$, and the inequality follows from the mixing property of the gossip matrix Property 2.6. Substituting the bound on $\mathbb{E}\|W_{t+\frac{1}{2}} - \bar{W}_{t+\frac{1}{2}}\|_F^2$ in Equation (23) then yields:

$$M\Xi_{t+1} \leq (1 - \rho)\left(1 + \frac{1}{\gamma}\right)M\Xi_t + (1 - \rho)(1 + \gamma)\eta^2\alpha_t^2 M\Psi_t .$$

Dividing by $M$ and setting $\gamma = 2/\rho$ finally gives:

$$\Xi_{t+1} \leq (1 - \rho)\left(1 + \frac{\rho}{2}\right)\Xi_t + (1 - \rho)\left(1 + \frac{2}{\rho}\right)\eta^2\alpha_t^2\Psi_t \leq \left(1 - \frac{\rho}{2}\right)\Xi_t + \frac{2\eta^2\alpha_t^2}{\rho}\Psi_t ,$$

where in the last inequality we used $(1 - \rho)(1 + \frac{\rho}{2}) = 1 - \frac{\rho}{2} - \frac{\rho^2}{2} \leq 1 - \frac{\rho}{2}$ and $(1 - \rho)(1 + \frac{2}{\rho}) = -1 - \rho + \frac{2}{\rho} \leq \frac{2}{\rho}$. $\square$

Using this recursion, we derive an explicit bound on $\Xi_t$ as follows.

**Lemma C.5.** *For any non-decreasing sequence $\{\alpha_t\}_{t \geq 1}$, the consensus distance $\Xi_t$ is bounded as:*

$$\Xi_t \leq \frac{2\eta^2\alpha_t^2}{\rho}\sum_{\tau=1}^{t-1}\left(1 - \frac{\rho}{2}\right)^{t-1-\tau}\Psi_\tau$$

*Proof.* The recursion for $\Xi_t$ in Lemma C.4 exhibits the structure in Lemma D.4. Applying the lemma with $a_t = \Xi_t$, $b_t = \eta^2\alpha_t^2\Psi_t$, and $\kappa = \rho/2$, we obtain:

$$\Xi_t \leq \frac{2}{\rho}\sum_{\tau=1}^{t-1}\left(1 - \frac{\rho}{2}\right)^{t-1-\tau}\eta^2\alpha_\tau^2\Psi_\tau \leq \frac{2\eta^2\alpha_t^2}{\rho}\sum_{\tau=1}^{t-1}\left(1 - \frac{\rho}{2}\right)^{t-1-\tau}\Psi_\tau ,$$

where in the last inequality we used $\alpha_\tau^2 \leq \alpha_t^2$ for all $\tau \leq t$. $\square$

## D. Technical Lemmas

In this section, we list some technical lemmas, starting with the following property that establishes the invariance of gossip communication when all machines hold the same vector.

**Lemma D.1** (Proposition 1, Koloskova et al., 2020)**.** *Let $x \in \mathbb{R}^d$. For any matrix $X = \begin{pmatrix} x & x & \cdots & x \end{pmatrix} \in \mathbb{R}^{d \times M}$ and a symmetric, doubly stochastic matrix $P \in \mathbb{R}^{M \times M}$, we have $XP = X$.*

Next, we state a classical result for smooth functions.

**Lemma D.2.** *Let $f : \mathbb{R}^d \to \mathbb{R}$ be an $L$-smooth function and $x^* \in \arg\min_{x \in \mathbb{R}^d} f(x)$. Then, for every $x \in \mathbb{R}^d$, it holds that:*

$$\|\nabla f(x)\|^2 \leq 2L \left(f(x) - f(x^*)\right) .$$

The following lemma proves to be useful as well.

**Lemma D.3.** *Consider a non-negative sequence $a_1, \ldots, a_T$ satisfying the following relation for any $t \in [T]$:*

$$a_t \leq b + \frac{1}{2T} \sum_{\tau=1}^{T} a_\tau ,$$

*for some constant $b \in \mathbb{R}$. Then, for all $t \in [T]$, it holds that*

$$a_t \leq 2b .$$

*Proof.* Summing over $t \in [T]$, we have that:

$$\sum_{t=1}^{T} a_t \leq Tb + \frac{1}{2T} \sum_{t=1}^{T} \sum_{\tau=1}^{T} a_\tau = Tb + \frac{1}{2} \sum_{t=1}^{T} a_t .$$

Subtracting $\frac{1}{2} \sum_{t=1}^{T} a_t$ and multiplying by 2 results in $\sum_{t=1}^{T} a_t \leq 2Tb$. Thus, we obtain:

$$a_t \leq b + \frac{1}{2T} \sum_{\tau=1}^{T} a_\tau \leq b + \frac{1}{2T} 2Tb = 2b .$$

$\square$

The following two lemmas are used to derive explicit bounds for the consensus recursions discussed in Appendix C.

**Lemma D.4.** *Let $\kappa \in (0, 1]$ and consider a sequence $\{a_t\}_{t \geq 1}$ satisfying the recursion:*

$$a_t \leq (1 - \kappa)a_{t-1} + \frac{b_{t-1}}{\kappa}, \quad \text{with} \quad a_1 = 0,$$

*for some non-negative sequence $\{b_t\}_{t \geq 1}$. Then, for all $t \geq 2$, the following bound holds:*

$$a_t \leq \frac{1}{\kappa} \sum_{\tau=1}^{t-1} (1 - \kappa)^{t-1-\tau} b_\tau .$$

*Proof.* We prove the statement using induction.

**Base case.** For $t = 2$, since $a_1 = 0$, it trivially follows that:

$$a_2 \leq (1 - \kappa)a_1 + \frac{b_1}{\kappa} = \frac{b_1}{\kappa} = \frac{1}{\kappa} \sum_{\tau=1}^{1} (1 - \kappa)^{1-\tau} b_\tau .$$

**Induction step.** Assume that the induction hypothesis holds for some $t \geq 2$. We show that it holds for index $t + 1$:

$$a_{t+1} \leq (1 - \kappa)a_t + \frac{b_t}{\kappa} \leq (1 - \kappa)\frac{1}{\kappa} \sum_{\tau=1}^{t-1} (1 - \kappa)^{t-1-\tau} b_\tau + \frac{b_t}{\kappa} = \frac{1}{\kappa} \left( \sum_{\tau=1}^{t-1} (1 - \kappa)^{t-\tau} b_\tau + b_t \right) = \frac{1}{\kappa} \sum_{\tau=1}^{t} (1 - \kappa)^{t-\tau} b_\tau .$$

$\square$

**Lemma D.5.** *Let $\kappa \in (0,1]$ and consider a sequence $\{a_t\}_{t \geq 1}$ satisfying the recursion:*

$$a_t \leq \left(1 - \kappa + \frac{c_1^2 \eta^2}{\kappa}\right) a_{t-1} + \frac{c_1^2 \eta^2}{\kappa^2} \sum_{\tau=1}^{t-2} (1-\kappa)^{t-2-\tau} a_\tau + \frac{c_2^2 \eta^2}{\kappa^3}, \quad \text{with} \quad a_1 = 0,$$

*for some non-negative constants $c_1$, $c_2$, and $\eta$. Suppose $\eta \leq \frac{\kappa^2}{2c_1}$. Then, for all $t \geq 1$, the following bound holds:*

$$a_t \leq \frac{2c_2^2 \eta^2}{\kappa^4}.$$

*Proof.* Similar to Lemma D.4, we prove this result by (strong) induction.

**Base cases.** For $t = 1$, the statement is trivial because $a_1 = 0 \leq 2c_2^2 \eta^2 / \kappa^4$. For $t = 2$, we have:

$$a_2 \leq \left(1 - \kappa + \frac{c_1^2 \eta^2}{\kappa}\right) a_1 + \frac{c_1^2 \eta^2}{\kappa^2} \sum_{\tau=1}^{0} (1-\kappa)^{-\tau} a_\tau + \frac{c_2^2 \eta^2}{\kappa^3} = \frac{c_2^2 \eta^2}{\kappa^3} \leq \frac{2c_2^2 \eta^2}{\kappa^4},$$

where the last inequality follows from $1 \leq \frac{2}{\kappa}$ as $\kappa \leq 1$, thus verifying the base cases.

**Induction step.** Suppose that for some $t \geq 3$, the induction hypothesis holds for all indices $s = 1, \ldots, t$. We shall prove that it then holds for $t + 1$. Specifically, denoting the upper bound by $B := \frac{2c_2^2 \eta^2}{\kappa^4}$, we assume that $a_s \leq B$ for $s = 1, \ldots, t$ and prove that $a_{t+1} \leq B$. Plugging the induction hypothesis into the recursion, we get:

$$a_{t+1} \leq \left(1 - \kappa + \frac{c_1^2 \eta^2}{\kappa}\right) a_t + \frac{c_1^2 \eta^2}{\kappa^2} \sum_{\tau=1}^{t-1} (1-\kappa)^{t-1-\tau} a_\tau + \frac{c_2^2 \eta^2}{\kappa^3}$$

$$\leq \left(1 - \kappa + \frac{c_1^2 \eta^2}{\kappa}\right) B + \frac{c_1^2 \eta^2}{\kappa^2} \sum_{\tau=1}^{t-1} (1-\kappa)^{t-1-\tau} B + \frac{c_2^2 \eta^2}{\kappa^3}$$

$$= \left(1 - \kappa + \frac{c_1^2 \eta^2}{\kappa} + \frac{c_1^2 \eta^2}{\kappa^2} \sum_{\tau=1}^{t-1} (1-\kappa)^{t-1-\tau}\right) B + \frac{c_2^2 \eta^2}{\kappa^3}$$

$$\leq \left(1 - \kappa + \frac{c_1^2 \eta^2}{\kappa} + \frac{c_1^2 \eta^2}{\kappa^3}\right) B + \frac{c_2^2 \eta^2}{\kappa^3}$$

$$\leq \left(1 - \kappa + \frac{2c_1^2 \eta^2}{\kappa^3}\right) B + \frac{c_2^2 \eta^2}{\kappa^3},$$

where the last inequality holds because $\frac{1}{\kappa} \leq \frac{1}{\kappa^3}$ for any $\kappa \leq 1$, and the penultimate inequality results from the bound on the geometric series:

$$\sum_{\tau=1}^{t-1} (1-\kappa)^{t-1-\tau} = \sum_{s=0}^{t-2} (1-\kappa)^s \leq \sum_{s=0}^{\infty} (1-\kappa)^s = \frac{1}{\kappa}.$$

From the condition on $\eta$, it follows that $\frac{2c_1^2 \eta^2}{\kappa^3} \leq \frac{\kappa}{2}$. Substituting this inequality and noting that $\frac{c_2^2 \eta^2}{\kappa^3} = \frac{\kappa}{2} B$, we obtain:

$$a_{t+1} \leq \left(1 - \frac{\kappa}{2}\right) B + \frac{c_2^2 \eta^2}{\kappa^3} = \left(1 - \frac{\kappa}{2}\right) B + \frac{\kappa}{2} B = B,$$

thus establishing the result. $\square$

We also utilize the following result, which we prove using simple algebraic manipulation.

**Lemma D.6.** *Let $A \geq 0$ and $B, C > 0$, and define $\eta = \min\left\{\eta_1, \sqrt{A/B}, (A/C)^{1/4}\right\}$ for some $\eta_1 > 0$. Then,*

$$\frac{A}{\eta} + B\eta + C\eta^3 \leq \frac{A}{\eta_1} + 2\sqrt{AB} + 2A^{3/4}C^{1/4}.$$

*Proof.* Since $\eta$ is the minimum between three terms, $1/\eta$ is the maximum of their inverses. Therefore,

$$
\begin{aligned}
\frac{A}{\eta} + B\eta + C\eta^3 &\le A \max \left\{ \frac{1}{\eta_1}, \sqrt{\frac{B}{A}}, \left(\frac{C}{A}\right)^{1/4} \right\} + B\eta + C\eta^3 \\
&\le A \left( \frac{1}{\eta_1} + \sqrt{\frac{B}{A}} + \left(\frac{C}{A}\right)^{1/4} \right) + B\sqrt{\frac{A}{B}} + C \left( \frac{A^{1/4}}{C^{1/4}} \right)^3 \\
&= \frac{A}{\eta_1} + \sqrt{AB} + A^{3/4}C^{1/4} + \sqrt{AB} + A^{3/4}C^{1/4} \\
&= \frac{A}{\eta_1} + 2\sqrt{AB} + 2A^{3/4}C^{1/4} \;,
\end{aligned}
$$

where the second inequality follows from the fact that the maximum of non-negative terms is smaller than their sum, and from the monotonic increase of the terms $B\eta$ and $C\eta^3$ with $\eta$. $\qquad\square$

## E. Additional Experimental Results

In Figures 4 to 6, we present the full convergence curves for DAT-SGD and D-SGD on the synthetic least squares problem over the ring, torus, and 1-peer exponential graph topologies, respectively. These results are shown for varying numbers of machines and correspond to the final errors reported in Figure 1.

On the torus and exponential graph topologies, our method steadily improves with increasing numbers of machines, demonstrating the potential of our approach to effectively increase parallelism. For the ring topology, performance improves when transitioning from $M = 9$ to 25 machines but degrades at $M = 100$ machines, as the topology-related error term becomes dominant. Conversely, D-SGD does not improve when increasing the number of machines. On the ring and torus topologies, performance is initially better for larger $M$, but then deteriorates, indicating that while we initially benefit from variance reduction, the optimization transitions to a regime where the leading error term depends on topology and inefficient information flow hinders further improvement. For the well-connected exponential graph, initial performance improves with $M$, but all configurations eventually converge to approximately the same error level.

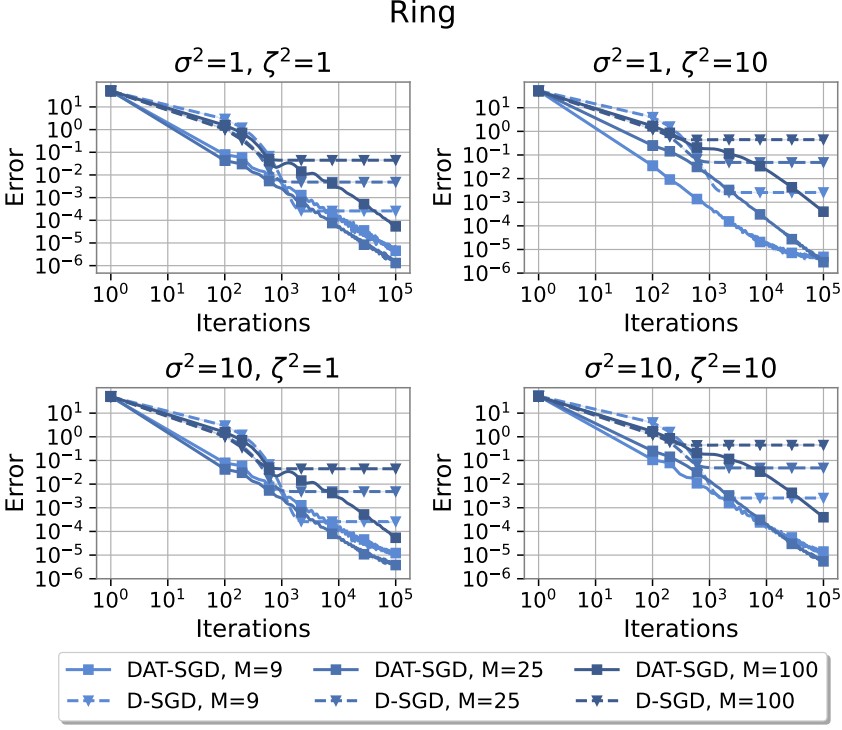

*Figure 4.* Error curves for the least squares problem with varying noise, data heterogeneity, and # of machines in the **ring** topology.

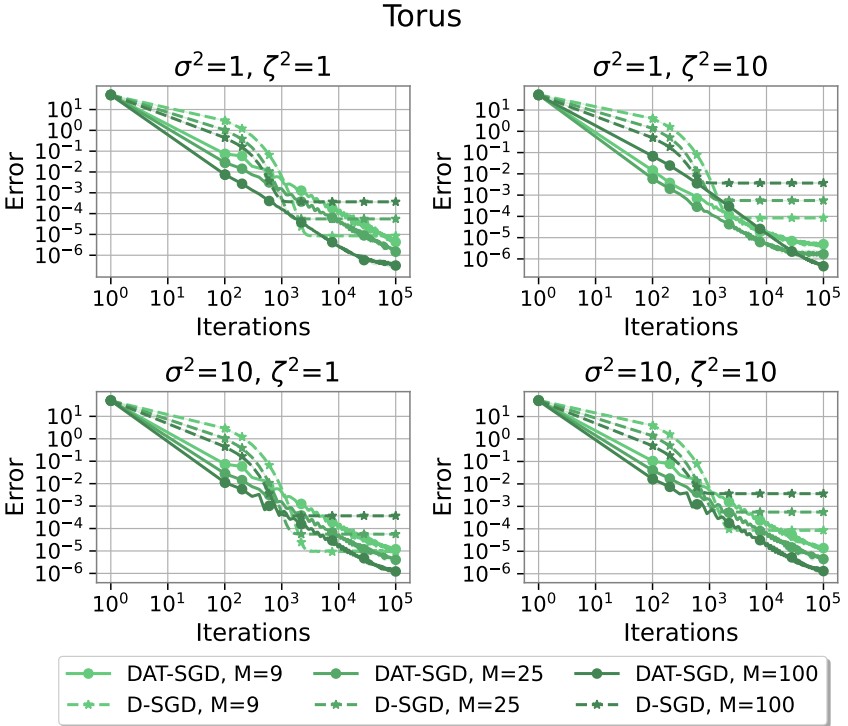

*Figure 5.* Error curves for the least squares problem with varying noise, data heterogeneity, and # of machines in the **torus** topology.

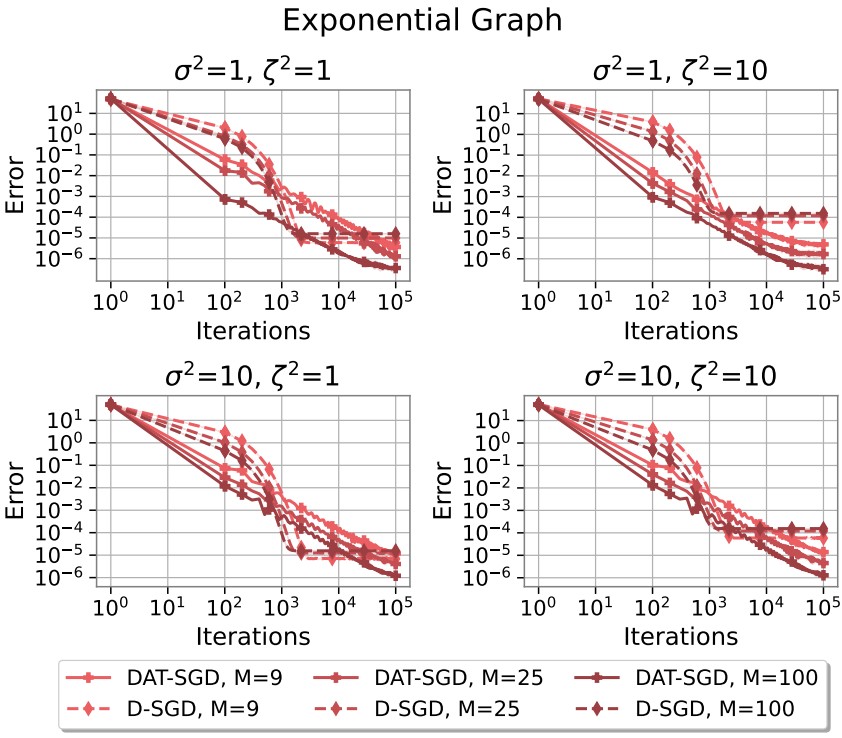

*Figure 6.* Error curves for the least squares problem with varying noise, data heterogeneity, and # of machines in the **1-peer exponential graph** topology.

