# OpenReview forum: "Enhancing Parallelism in Decentralized Stochastic Convex Optimization"
_ICML.cc/2025/Conference — ICML 2025 poster_

### Official Review · Reviewer_X6LA · 2025-02-27

**Overall Recommendation:** 3

**Summary:**

This paper presents Decentralized Anytime SGD, a decentralization optimization algorithms that is based on Anytime SGD. The authors presents the convergence analysis of Decentralized Anytime SGD. Decentralized Anytime SGD achieves linear speedup and has a better sample complexity than that of D-SGD under the non-convex scenario.

**Claims And Evidence:**

Yes.

**Essential References Not Discussed:**

NA

**Experimental Designs Or Analyses:**

There is no experiments in this paper. However, as a decentralized optimization algorithm, it is essential to present numerical experiments to compare the proposed algorithm with other classial decentralized algorithms (including D-SGD and more), which can validate the practical performance of the proposed algorithms.

**Methods And Evaluation Criteria:**

Yes.

**Other Comments Or Suggestions:**

Linear speedup is a significant character of decentralized algorithms. Although the proposed algorithm achieves the linear speedup, the authors fail to high it and use the sample complexity $N$ in the convergence result. The reviewer suggest the author to add more discussion on linear speedup and highlight it in the theoretical result.

**Other Strengths And Weaknesses:**

**Strengthes**
1. Present a proof sketch of the convergence rate.
2. Achieves linear speedup.

**Weakness**
1. Lack of the analysis of transit complexity.

**Questions For Authors:**

1. Whether the authors can present additional experiments to compare the proposed algorithm to other decentralized algorithms?
2. Can the author provide any discussion on linear speedup and transient complexity?
3. Whether the symmetric assumption of the gossip matrix $P$ can reduce? Why?

The reviewer would to update the final rating according to the response to those weaknesses and questions, as well as the experimental performance.

**Relation To Broader Scientific Literature:**

NA

**Theoretical Claims:**

The reviewer has check the theoretical claims and proofs in this paper and has not found any fatal issues.

---

> ### Author Rebuttal · Authors · 2025-03-31
>
> We thank the reviewer for their time and feedback. We address the reviewer’s questions individually:
>
> **Experiments**: We have included experiments evaluating our method on both a synthetic, convex least squares problem and non-convex neural network training. We refer the reviewer to our response to **Reviewer 3nY6** for a discussion of the results.
>
> **Linear speedup and transient time**: We will add a discussion about the transient complexity in the revised version. It can be inferred from Theorem 4.1 that the transient time for our method is $T\geq\mathcal{O}(M/\rho^2)$; this improves upon D-SGD by a factor of $M^2$. For example, this implies a transient complexity of $\mathcal{O}(M^5)$ for a ring and $\tilde{\mathcal{O}}(M)$ for a static exponential graph. Should the reviewer have any further comparisons in mind, we would be happy to incorporate them into our text.
>
> **Symmetric gossip matrix assumption**: Thank you for raising this insightful point. Our analysis relies on the contraction property stated in Property 2.6 (Eq. (2)), which holds when the communication matrix is symmetric and doubly stochastic. This assumption is standard in the literature and has been used in many prior works, including [1,2,3,4]. We acknowledge that there is a growing body of work analyzing more general communication matrices—such as asymmetric and/or only row-/column-stochastic matrices—e.g., [5,6,7]. Our goal in this work was to provide a clean and interpretable analysis under a widely adopted and well-studied assumption. We believe our results open the door to extending the analysis to more general settings with non-symmetric or non-doubly-stochastic matrices. We will include a discussion of this direction in the 'Conclusion and Future Work' section.
>
> [1] Lian et al., “Can decentralized algorithms outperform centralized algorithms? a case study for decentralized parallel stochastic gradient descent”, '17
> [2] Tang et al., “Communication compression for decentralized training”, '18
> [3] Koloskova et al., “A unified theory of decentralized sgd with changing topology and local updates”, '20
> [4] Koloskova et al., “An improved analysis of gradient tracking for decentralized machine learning”, '21
> [5] Assran et al., “Stochastic gradient push for distributed deep learning”, '19
> [6] Pu & Nedic. “Distributed stochastic gradient tracking methods”, '21
> [7] Lu & De Sa. “Optimal Complexity in Decentralized Training”, '21

---

> > ### Comment · Reviewer_X6LA · 2025-04-02
> >
> > The reviewer has no additional problems and decides to update the rating.

---

> > > ### Author Response · Authors · 2025-04-02
> > >
> > > We sincerely thank the reviewer for positively updating their evaluation and for their constructive feedback, which helped us further strengthen our paper.

---

### Official Review · Reviewer_75sX · 2025-03-14

**Overall Recommendation:** 2

**Summary:**

The paper proposes Decentralized Anytime SGD — a noved algorithm for decentralized optimization. The algorithm is based on anytime SGD algorithm proposed by (Cutkosky, 2019). The paper provides the convergence rate of their method for convex functions, showing improvement over D-SGD in the middle convergence term, as well as showing the convergence for the last iterate averaged across the nodes, instead of the priorly used average of losses from all the iterates.

## update after rebuttal

I would like to thank the authors for their response and for adding the experiments. While some of my concerns have cleared, I have some of the remaining concerns and therefore I keep my score.
- I disagree with the authors that their method always improves over the baselines theoretically. For example, when data heterogeniety term is large, then Gradient Tracking is expected to have the better convergence (see Table 2 in this submission). Even with this, I believe that the paper provides an interesting improvements over the existing decentralized methods in the homogeneous case, however, I would like to see a more rigorous discussion of this.
- Given that, I am a bit surprised that experiments show the opposite from theory: that D-SGD and D^2 improve in the homogeneous case, while DAT-SGD improves in the heterogeneous case.
- I beleive that the Gradient Tracking method is not orthogonal, but a direct baseline mehtod, and therefore it should be included in the experimental comparison. For example, in the theoretical comparison in Table 2 of this submission, the proposed algortihm DAT-SGD was combared with Gradient Tracking but was not compared with D^2. Thus, I do not understand the choice of the baseline of D^2 instead of Gradient Tracking in experiments.
- For tuning hyperparameters in experimental comparison, please ensure that the optimal learning rate is not on the end of the grid by extending the grid when necessary. E.g. in neral network experiments the learning rate was chosen only from 3 values, which makes it very likely that for some experiments, the found learning rate was on the end of the tuned grid.

**Claims And Evidence:**

The algorithm is interesting and the proposed algorithm provides an interesting and non-trivial improvement in some cases over the prior decentralized SGD algorithms for the convex smooth functions.

**Essential References Not Discussed:**

-

**Experimental Designs Or Analyses:**

How does the proposed algoritm compares to D-SGD and GT emperically? Can we see in practice the benefit of the improved convergence rate?

**Methods And Evaluation Criteria:**

theoretical part yes, however there is no empirical evaluation of the proposed algorithm.

**Other Comments Or Suggestions:**

-

**Other Strengths And Weaknesses:**

The method is limited to the convex functions only

**Questions For Authors:**

Could you characterize, under which conditions the proposed method improves over the prior works? I.e. what are the conditions on \rho and \zeta, under which the proposed algorithm improves convergence? Also, e.g. could highlight the case of the homogeneous function with zeta=0 and give a condition on rho.

Can the proposed method be generalized for non-convex smooth functions?

**Relation To Broader Scientific Literature:**

Has there been any other work on decentralized optimization methods showing last-iterate convergence?

Has there been any lower bounds for the convex decentralized optimization? How does the provided convergence rate compare to those lower bounds?

**Theoretical Claims:**

I am not sure of correctness of the proofs since when I started to check the proof, already statement of Lemma A.1. seems to have a typo: it should be alpha_{tau - 1} instead of alpha_{tau}. Moreover while checking the proof of Theorem 1 from (Dahan & Levy), I noticed that it uses iterates x_{tau - 1} for tau = 0, however x_{-1} was never defined. Please clarify these points, as right now the proof seem to be incorrect.

---

> ### Author Rebuttal · Authors · 2025-03-31
>
> We thank the reviewer for their time and valuable input. We address the reviewer’s concerns and questions separately:
>
> **Correctness of the proof and Lemma A.1**: We divide our answer into 2 parts:
> - First, the reviewer’s concern about the appearance of the term $x_{-1}$ in the analysis of [1] refers specifically to the equality $\alpha_{\tau}(x_{\tau}-w_{\tau}) = \alpha_{0:\tau-1}(x_{\tau-1} - x_{\tau})$, which is applied for $\tau=0,...,T$ (in Appendix D therein). Although the authors of [1] did not explicitly state this, by definition we have $\alpha_{0:-1}=0$ (also see the first line in the proof of Theorem 1 in [2]; they start at $t=1$ like we do, thus defining $\alpha_{1:0}=0$). Therefore, at $\tau=0$, both sides of this equality trivially evaluate to zero, since $w_{0}=x_{0}$ and $\alpha_{0:-1}=0$. Consequently, despite the formal appearance of the term $x_{-1}$, it can be defined arbitrarily since it is multiplied by zero; thus, the equality (and therefore Theorem 1 in [1]) remains valid.
> - Second, regarding Lemma A.1, we thank the reviewer for helping us spot a typo; however, we clarify that the typo does not occur within Lemma A.1 itself and does not impact our results in any way. Nevertheless, the typo requires slight adjustments to the text, as we elaborate below. The error appears in Eq. (6) (and similarly in Eq. (16)), where the coefficients should be corrected to: $x_{t+1}=\frac{\alpha_{1:t}}{\alpha_{1:{t+1}}}x_{t}+\frac{\alpha_{{t+1}}}{\alpha_{1:{t+1}}}w_{t+1}$.
>    - With this correction, our Lemma A.1 aligns precisely with Theorem 1 in [1], except for the indexing of iterations—[1] starts indexing from $t=0$, while we start from $t=1$. Therefore, all summations in our analysis (including terms like $\alpha_{1:t}$) begin from $\tau=1$ instead of $\tau=0$. After this correction, Lemma A.1 is accurate and correctly stated in its current form.
>    - The typo correction also necessitates a minor update in the definition of $\delta_t$ at Line 594: it should now be $\delta_{t}=\alpha_{t+1}/\alpha_{1:t+1}$ instead of $\alpha_{t}/\alpha_{1:t}$. Importantly, this adjustment does not affect Lemma C.3, since for $\alpha_{t}=t$, we now have $\delta_{t} = 2/(t+2)$ and consequently $\alpha_{t}\delta_{t}=2t/(t+2)$, which still satisfies the condition $\alpha_{t}\delta_{t}\leq 2$ (Lines 903-904).
>    - The only additional proof requiring modification is Lemma B.1, which remains correct once these coefficients are appropriately updated.
>
> We hope this clarification resolves any potential confusion.
>
> **Experiments**: We have added experimental results and refer the reviewer to our response to **Reviewer 3nY6** for further discussion. In the experiments, we compare our method with both D-SGD and $D^2$ (for the non-convex image classification task; as suggested by **Reviewer 3nY6**), with the latter being “more tolerant to data heterogeneity”. We note that Gradient Tracking (GT) is orthogonal to our method; the tracking mechanism can also be applied to our approach by tracking the gradients at the query points $x_t^i​$. Investigating the effect of GT, both theoretically and practically, is a valuable future direction.
>
> **Last-iterate convergence**: To the best of our knowledge, there is no other work in the decentralized setup showing last-iterate convergence.
>
> **Lower bounds for decentralized SCO**: The first statistical term, of order $\sigma/\sqrt{MT}$, matches the centralized rate and is unimprovable; see, e.g., [3,4]. While we are not aware of any lower bound for the second term (of order $1/\rho T$ in our rate), which is related to the network topology, our derived parallelism bound of $M\leq\mathcal{O}(\rho\sqrt{N})$ is unimprovable in terms of $N$ (i.e., $\sqrt{N}$), as it matches the centralized case. It remains an interesting open question whether the dependence on $\rho$ can be further improved.
>
> **Convex analysis**: The reviewer is correct – our analysis is valid for convex functions. We have included experiments on neural network training to demonstrate our method's potential in non-convex optimization scenarios. Establishing convergence bounds for non-convex functions is a non-trivial task we leave for future work.
>
> **Improvement w.r.t prior work**: Our proposed method improves over prior work for any value of $\rho$. Note that the second term in our rate, of order $1/\rho T$, also appears in the convergence bounds of D-SGD and GT. However, our analysis removes the term that scales with $1/\rho^{1/3}T^{2/3}$, which limits the achievable parallelism bound.
>
> [1] Dahan & Levy, “SLowcal-SGD: Slow Query Points Improve Local-SGD for Stochastic Convex Optimization”, '24
> [2] Cutkosky, “Anytime Online-to-Batch, Optimism and Acceleration”, '19
> [3] Woodworth et al., “Graph oracle models, lower bounds, and gaps for parallel stochastic optimization”, '18
> [4] Woodworth et al., “The Min-Max Complexity of Distributed Stochastic Convex Optimization with Intermittent Communication”, '21

---

### Official Review · Reviewer_ejda · 2025-03-14

**Overall Recommendation:** 2

**Summary:**

The paper introduces Decentralized Anytime SGD (DAT-SGD) to enhance parallelism in decentralized stochastic convex optimization (SCO).

Main Findings
DAT-SGD extends the parallelism threshold to O(ρ√N), matching centralized learning, while prior decentralized methods were limited to O(ρ¹/²N¹/⁴).

Main Results
The algorithm achieves an improved error bound of O(σ/√MT + (√σ+√ζ)/ρT + 1/T), enabling efficient large-scale decentralized training.

Main Algorithmic Idea
DAT-SGD builds on Anytime SGD, using averaged iterates and gossip averaging to reduce consensus bias, improving convergence and scalability in decentralized networks.

**Claims And Evidence:**

The paper’s claims are well-supported by rigorous theoretical analysis and comparisons with prior work. It effectively demonstrates that DAT-SGD improves parallelism to O(ρ√N), surpassing previous decentralized methods. The claim that DAT-SGD mitigates consensus distance is backed by mathematical proofs, showing reduced model divergence through averaged iterates. Additionally, the improved error bound provides strong evidence for faster and more stable convergence.

However, the paper lacks empirical validation. Experimental results comparing DAT-SGD with existing decentralized methods would further strengthen the claims and demonstrate its real-world applicability.

**Essential References Not Discussed:**

NA

**Experimental Designs Or Analyses:**

The paper does not include an experimental section, making it difficult to validate the practical effectiveness of DAT-SGD.

**Methods And Evaluation Criteria:**

The proposed DAT-SGD method is well-grounded in theoretical analysis, focusing on improving parallelism in decentralized stochastic convex optimization. The convergence bounds and parallelism limits provide strong analytical validation. However, the paper lacks empirical evaluation, which is crucial for assessing real-world performance. Including experiments on benchmark datasets and various network topologies would strengthen the evaluation. While the theoretical framework is solid, practical testing would provide a more comprehensive understanding of scalability, robustness, and efficiency in real decentralized learning environments

**Other Comments Or Suggestions:**

NA

**Other Strengths And Weaknesses:**

The primary weakness of the paper is the lack of an experimental section, making it difficult to assess the practical effectiveness of DAT-SGD. Without empirical validation, it remains unclear how the method performs in real-world decentralized learning scenarios. Benchmark experiments on different network topologies and datasets would significantly strengthen the paper.

**Questions For Authors:**

Do you plan to include experiments in an extended version or supplementary material?
If there are ongoing or planned experiments, providing preliminary results or an outline of the experimental setup would help in evaluating the method’s real-world applicability.

**Relation To Broader Scientific Literature:**

This paper builds on two key works: Koloskova et al. (2020) and Cutkosky (2019). Koloskova et al. (2020) developed a unified theory for decentralized SGD, addressing topology changes and local updates but with limited parallelism scalability. DAT-SGD improves upon this by enhancing parallelism to O(ρ√N) while maintaining strong convergence guarantees.

Cutkosky (2019) introduced Anytime SGD, which leverages averaged iterates for stable updates. The authors extend this idea to decentralized settings, using it to mitigate consensus distance and improve statistical efficiency.

**Theoretical Claims:**

The proofs in Sections 4.3 and 4.4 were reviewed, particularly the proof sketches outlining the convergence analysis and bias reduction in DAT-SGD. The arguments appear logically sound, with a clear derivation of key results such as the improved parallelism bound O(ρ√N). The analysis effectively builds on Anytime SGD and consensus distance reduction. No major issues were found

---

> ### Author Rebuttal · Authors · 2025-03-31
>
> We appreciate the reviewer’s constructive feedback. It appears that the reviewer’s primary concern is the lack of experimental results. In response, we have included experiments evaluating our method on both a synthetic convex problem and non-convex neural network training. We refer the reviewer to our response to **Reviewer 3nY6** for a detailed discussion of the results.

---

### Official Review · Reviewer_3nY6 · 2025-03-14

**Overall Recommendation:** 4

**Summary:**

The paper studies an anytime variant of decentralized SGD. It achieves bounds allowing a larger number of nodes successfully team up in decentralized training. It does so by using gradients at averaged query points, thus improving the consensus distance and thus convergence under large number of nodes, which is a valuable contribution.

**Claims And Evidence:**

The paper improves the convergence results for decentralized SGD, and also gives a last-iterate convergence result, both being interesting additions to the understanding of such methods.

The algorithmic change is simple & elegant and has no implementation downsides, yet leads to the significantly improved convergence results mentioned.

**Essential References Not Discussed:**

-

**Experimental Designs Or Analyses:**

No experimental results are provided unfortuantely.

I know this will sound as a typical ICML reviewer cliche, but I think here some experiments would improve the value of the paper. The claim of higher tolerance to larger number of nodes is very clear, well-supported by theory, so it would really add value to the work to verify this phenomenon on large graphs on simple settings, with the competing methods that you already theoretically compare. In addition, several methods which are more tolerant to data heterogeneity could be included in the comparison (e.g. D^2).

This would not be very hard to do as many good codebases simulating DSGD are available by now.

UPDATE: I appreciate that the authors have added experiments along the main research aspects now, which I think adds value to the paper.

Given this, i have upgraded my rating to 'accept'.

**Methods And Evaluation Criteria:**

Yes, this is a theory paper

**Other Comments Or Suggestions:**

-

**Other Strengths And Weaknesses:**

-

**Questions For Authors:**

see above

**Relation To Broader Scientific Literature:**

Looks appropriate

**Theoretical Claims:**

See claims & evidence above

In terms of presentation, the proof sketch was very useful. I could not check the full proof in detail but overall the approach looks plausible and appropriate

---

> ### Author Rebuttal · Authors · 2025-03-31
>
> We thank the reviewer for acknowledging our contributions and for the positive feedback. As suggested, we provide experiments to evaluate our method, including a synthetic least squares problem and an image classification task. All experiments are run with 3 random seeds, and we report the average performance. Results are available in the following anonymized GitHub repo; we recommend downloading the repo for optimal examination: https://anonymous.4open.science/r/DAT-SGD-figures-4378
>
> **Synthetic Least Squares**: For machine $i\in[1,...,M]$, the local objective is $f_i(x)=\frac{1}{2}\lVert A_ix-b_i\rVert^2$, with $A_i\in\mathbb{R}^{d\times{d}}$ drawn from $\mathcal{N}(0,I)$. The targets are given by $b_i=A_i(x^\sharp-\delta_i)$, where $x^\sharp\sim\mathcal{N}(0,I/d)$ is sampled once per run and $\delta_i\sim\mathcal{N}(0,\zeta^2 I/d)$ introduces heterogeneity. To incorporate stochasticity, we add noise $\xi\sim\mathcal{N}(0,\sigma^2 I/d)$, yielding the noisy gradient $\nabla f_i(x)+\xi$.
>
> We compare our method with D-SGD over ring, torus, and exponential graph (for which $1/\rho=\mathcal{O}(\log{M})$) topologies, varying $\sigma,\zeta\in[1,10]$, number of machines $M\in[4,9,25,49,100]$, and $d=50$. For each run, we tuned the learning rate via a grid search over $\eta\in[0.1,5e-2,1e-2,5e-3,1e-3,5e-4,1e-4]$.
>
> In ‘least_squares_parallelization.pdf’, we plot final errors ($\frac{1}{M}\sum_{i=1}^{M}{\lVert x_T^i - x^*\rVert^2}$ and $\frac{1}{M}\sum_{i=1}^{M}{\lVert w_T^i - x^*\rVert^2}$ for DAT-SGD and D-SGD, respectively) vs the number of machines for varying $\sigma$ and $\zeta$ and across topologies. Colors represent different topologies; line-style denotes the method (solid:DAT-SGD, dashed:D-SGD). For D-SGD, performance deteriorates as $M$ increases, and more significantly for less connected graphs (lower $\rho$). For ring, this degradation occurs from $M=4$, while for torus and exponential topologies performance is flat between $M=4$ and $M=9$ and degrades afterwards. In contrast, our method improves as $M$ grows: for torus and exponential topologies performance steadily improves, while for ring, it improves up to $M=25$ before deteriorating in a trend similar to that of D-SGD. This suggests that for some $M$ between 25 and 49, the network-related convergence term ($1/\rho T=\mathcal{O}(M^2/T)$ for ring) becomes dominant. Overall, this figure aligns with our theoretical findings: DAT-SGD enables performance improvement for larger $M$. We provide the complete convergence curves for different topologies and $M\in[9, 25,100]$ in ‘least_squares_curves_X.pdf’, where X denotes the topology (ring/torus/exponential).
>
> **Image Classification with a Neural Network**: We conduct experiments on Fashion MNIST using LeNet, comparing DAT-SGD with D-SGD and $D^2$ [1]. For DAT-SGD and D-SGD, we use momentum with $\beta=0.9$. Data is distributed among machines using a Dirichlet distribution with parameter $\alpha$ to control heterogeneity [2]. Experiments are performed on both a ring topology and the Base-2 Graph [3]-a time-varying, state-of-the-art topology for decentralized learning. Learning rates are tuned over $\eta\in[0.1,0.01,0.001]$. Colors represent methods and line styles indicate topology (solid:ring, dashed:Base-2).
>
> Unlike the convex least squares setting, this task is non-convex. Following the heuristic proposed by [4], we adopt a momentum-like Anytime update: $x_{t+1}=\gamma_t x_t+(1-\gamma_t)w_t$, with a fixed $\gamma_t=0.9$. Their work shows this enhances training stability and adaptability in non-convex landscapes.
>
> In ‘fashion_mnist_parallelization.pdf’, we plot final accuracy after 200 epochs vs $M$ for ring topology with heterogeneous data ($\alpha=0.1$). Our method outperforms the baselines; in addition, the largest accuracy drop for DAT-SGD occurs between $M=8$ and $M=16$, while D-SGD and $D^2$ degrade most between $M=4$ and $M=8$, demonstrating our increased parallelism claim.
>
> In ‘fashion_mnist_curves.pdf’, we show test accuracy vs epochs for $M\in[8,16]$ and $\alpha\in[0.1,10]$ (heterogeneous and homogeneous setups). In the heterogeneous case, our method outperforms baselines consistently over both topologies, with ring topology performance matching ($M=8$) or nearly matching ($M=16$) the baselines on the well-connected Base-2 graph. Conversely, in the homogeneous case, D-SGD and $D^2$ achieve better performance, motivating further study of our anytime averaging heuristic in non-convex scenarios.
>
> [1] Tang et al., “D^2: decentralized training over decentralized data”, ‘18
> [2] Hsu et al., “Measuring the effects of non-identical data distribution for federated visual classification”, ‘19
> [3] Takezawa et al., “Beyond exponential graph: Communication-efficient topologies for decentralized learning via finite-time convergence”, ‘23
> [4] Dahan & Levy, “Do stochastic, feel noiseless: stable stochastic optimization via a double momentum mechanism”, ‘25

---

### Decision · Program_Chairs · 2025-05-01

**Decision:**

Accept (poster)

**Comment:**

The paper gives a tighter convergence analysis of decentralized SGD than prior works such as Koloskova et al. The key idea that enables this improvement is Anytime SGD, extending the framework proposed in Cutoksky et al, where the algorithm maintains a slowly moving average iteration in addition to local models at the nodes. This slowly moving iterate helps reduce the consensus gap and improve convergence bounds. Here are some points due to which the paper is on the borderline.
* The paper did not have empirical evaluations, however, the authors did show some results during the rebuttal phase which they plan to add to the paper. This convinced some reviewers to increase their scores.
* The analysis only applies to stochastic convex optimization
* The improvement is in the higher order terms of the convergence bound, not the dominant term.
* The practical applicability of the proposed algorithm is limited due to the need to maintain average iterates